# Functional-Group-Based Diffusion for Pocket-Specific Molecule Generation and Elaboration

**Haitao Lin**
Westlake University
linhaitao@westlake.edu.cn

**Yufei Huang**
Westlake University
huangyufei@westlake.edu.cn

**Odin Zhang**
Zhejiang University
haotianzhang@zju.edu.cn

**Lirong Wu**
Westlake University
wulirong@westlake.edu.cn

**Siyuan Li**
Westlake University
lisiyuan@westlake.edu.cn

**Zhiyuan Chen**
Deep Potential
chenzhiyuan@dp.tech

**Stan Z. Li** *
Westlake University
stan.zq.li@westlake.edu.cn

## Abstract

In recent years, AI-assisted drug design methods have been proposed to generate molecules given the pockets' structures of target proteins. Most of them are *atom-level-based* methods, which consider atoms as basic components and generate atom positions and types. In this way, however, it is hard to generate realistic fragments with complicated structures. To solve this, we propose D3FG, a *functional-group-based* diffusion model for pocket-specific molecule generation and elaboration. D3FG decomposes molecules into two categories of components: functional groups defined as rigid bodies and linkers as mass points. And the two kinds of components can together form complicated fragments that enhance ligand-protein interactions. To be specific, in the diffusion process, D3FG diffuses the data distribution of the positions, orientations, and types of the components into a prior distribution; In the generative process, the noise is gradually removed from the three variables by denoisers parameterized with designed equivariant graph neural networks. In the experiments, our method can generate molecules with more realistic 3D structures, competitive affinities toward the protein targets, and better drug properties. Besides, D3FG as a solution to a new task of molecule elaboration, could generate molecules with high affinities based on existing ligands and the hotspots of target proteins.

## 1 Introduction

The established notion in drug-target identification is that similar structures perform similar functions. This principle allows classical computer-aided drug design (CADD) to abstract protein-ligand interactions as pharmacophores, aligning similar functional groups and extracting pharmaceutical

---

*Corrsponding Author.

37th Conference on Neural Information Processing Systems (NeurIPS 2023).

information from these structures. Fitting suitable functional groups into the pharmacophores can enhance ligand-protein interactions, thus improving drug efficiency [1, 2].

Artificial intelligence has already achieved outstanding success in protein design [3–8] and thus led to a new round of attention in drug design focused on AI-assisted drug design (AIDD). A number of methods of AI-assisted drug design on pocket-specific molecule generation are emerging [9–15], thanks to developments in deep generative models [16–21] and graph neural networks (GNNs) [22–31]. These methods usually focus on atom-level generation, which first generates atom types and positions and then assembles atoms into molecules that can bind to the protein pockets. Although significant progress has been made, they are still weak in two aspects. For one thing, it is hard for them to generate realistic functional groups that contribute pharmacological effects to the target as classical CADD methods are able to. It is shown that generating benzene rings is uncommon compared with the reference molecules, not to mention some large functional groups with complex structure constraints such as purine, indole, etc (Table. 1 in Sec. 5.2). For another, trade-offs between efficiency and sufficiency of protein context place them in a dilemma. For example, TARGETDIFF [13] employs sufficient protein context, which disassembles the amino acids into atoms, but leads to inefficiency due to a large node number (412.14 on average) in GNN's message passing. DIFFSBDD [14] simplifies the representation of protein context, by only using $C_\alpha$'s positions and residue types, resulting in a reduction of GNNs' node number (68.10 on average) but the insufficiency of context information.

To address the above issues, we establish a **f**unctional-**g**roup-based **diff**usion model (D3FG), including the following contributions: **1. Method Novelty.** We denote the molecules' functional groups and proteins' amino acids as the same level's fragments, in which the intra-relative positions of atoms are fixed like rigid bodies, and represent single atoms as linkers. The positions and orientation of local structures and the atom type variables are generated gradually through denoising processes. The fragment-linker designation leads the binding systems to heterogeneous graphs, and thereby, two schemes are proposed as solutions, which achieve competitive performance in terms of molecule structures, binding affinity, and drug properties, and sufficiency in encoding protein context by employing more features. **2. Dataset Establishment.** Though the CrossDocked2020 [32] has been a widely-used dataset for evaluation methods' performance on the task, the analysis of the molecules' functional groups of it is missing. We deeply explore the details of the inter-relative positions and types of functional groups of the molecules and establish an extendable database of common functional groups. **3. New task.** Besides molecule generation, we propose molecule elaboration as another task that our model can fulfill. Fragment hotspot maps (FHM) [33, 34] are used to preprocess paired protein-molecule in CrossDocked2020 for the task. As a result, our model generates molecules with high binding affinity based on the reference.

## 2 Problem Statement

For a binding system composed of a protein-molecule pair (also called protein-ligand pair) as $\mathcal{C}$, which contains $N_{aa}$ amino acids of proteins and $N_{fg}$ functional groups and $N_{at}$ single atoms of molecules, we represent the index set of the molecule's single atoms as $\mathcal{I}_{at}$, functional groups as $\mathcal{I}_{fg}$, and the protein's amino acids as $\mathcal{I}_{aa}$, where $|\mathcal{I}_{fg}| = N_{fg}$, $|\mathcal{I}_{at}| = N_{at}$ and $|\mathcal{I}_{aa}| = N_{aa}$. Note that a molecule can be disassembled into functional groups and single atoms other than functional groups, which we also call linkers. For a protein, the amino acids can be represented by its type, $C_\alpha$ atom coordinate, and the orientation, denoted as $s_i \in \{1, \dots, 20\}$, $\boldsymbol{x}_i \in \mathbb{R}^3$, $\boldsymbol{O}_i \in \text{SO}(3)$, where $i \in \mathcal{I}_{aa}$. For a molecule, assuming there are $M_{fg}$ and $M_{at}$ possible types in total functional groups and linker atoms respectively, the functional groups can be represented as the three elements if the inter-relative positions are fixed, as $s_j$, $\boldsymbol{x}_j$ and $\boldsymbol{O}_j$, where $s_j \in \{21, \dots, 21 + M_{fg}\}$ is the type, $\boldsymbol{x}_j \in \mathbb{R}^3$ is the predefined center atom's coordinate, and $\boldsymbol{O}_j \in \text{SO}(3)$ can also be obtained in the same way as amino acids (See Sec. 5.1 and Appendix. D.) for $j \in \mathcal{I}_{fg}$; And its linkers can be represented as $s_k$, $\boldsymbol{x}_k$ and $\boldsymbol{O}_k$, with $s_k \in \{22 + M_{fg}, \dots, 22 + M_{fg} + M_{at}\}$, $k \in \mathcal{I}_{at}$ and $\boldsymbol{O}_k = \text{diag}\{1, 1, 1\} = \boldsymbol{I}$.

Therefore, $\mathcal{C} = \{(s_i, \boldsymbol{x}_i, \boldsymbol{O}_i)\}_{i=1}^{N_{aa}+N_{fg}+N_{at}}$ can be split into two sets as $\mathcal{C} = \mathcal{P} \cup \mathcal{M}$, where $\mathcal{P} = \{(s_i, \boldsymbol{x}_i, \boldsymbol{O}_i) : i \in \mathcal{I}_{aa}\}$ and $\mathcal{M} = \{(s_j, \boldsymbol{x}_j, \boldsymbol{O}_j) : j \in \mathcal{I}_{fg} \cup \mathcal{I}_{at}\}$ . For protein-specific molecule generation, our goal is to establish a probabilistic model to learn the distribution of molecules conditioned on the target proteins, *i.e.* $p(\mathcal{M}|\mathcal{P})$. In the following, we omit $i \in \mathcal{I}_{aa}$ and $j \in \mathcal{I}_{fg} \cup \mathcal{I}_{at}$ by default unless specified.

## 3   Related Work

**3D molecule generation.**   Previous methods on molecule generation fucus on 1D-smiles-string [35–38] or 2D-molecule-graph [39–43]. In recent years, more works concentrate on 3D molecule generation, thanks to fast development in equivariant graph neural networks [44–46] and generative models [16–21]. Molecular conformation generation aims to generate 3D structures of molecules with stability, given 2D molecule graph structure, [47–51]. Further, De novo molecular generation attempts to generate both 2D chemical formulas and 3D structures from scratch [52–54].

**Fragment-based drug design.**   Previously, works on fragment-based molecule generation are proposed. For example, JT-VAE [55] generates a tree-structured scaffold over chemical substructures and combines them into a 2D-molecule. PS-VAE [56] can automatically discover frequent principal subgraphs from the dataset, and assemble generated subgraphs as the final output molecule in 2D. Further, DEEPFRAG [57] predicts fragments conditioned on parents and the pockets, SQUID [58] generates molecules in a fragment level conditioned on molecule's shapes. FLAG[59] auto-regressively generates fragments as motifs based on the protein structures in 3D.

**Structure-based drug design.**   Success in 3D molecule generation and an increasing amount of available structural data of protein and molecules raises scientific interests in structure-based drug design (SBDD), which aims to generate both 2D molecule graphs and 3D structures conditioned on target protein structure as contextual constraints. For example, LIGAN [60] and 3DSBDD [12] are grid-based models which predict whether the grid points are occupied by specific atoms. By harnessing 3D equivariant graph neural networks, POCKET2MOL [9] and GRAPHBP [10] generate atoms auto-regressively and model the probability of the next atom's type as discrete categorical attribute and position as continuous geometry. FLAG[59] is also fragment-based, but still generates motifs in an auto-regressive way. Recently, utilizing the diffusion denoising probability models [61–63, 53, 64], a series of SBDD methods generate ligands conditioned on the target pockets at full atom levels [13–15].

## 4   Method

D3FG firstly decomposes molecules into two categories of components: functional groups and linkers, and them use the diffsion generative model to learn the type and geometry distributions of the components. In this section, we describe the D3FG by four parts: (i) the diffusion model as the generative framework, in which the three variables are generated; (ii) the denoiser parameterized by graph neural networks, satisfying certain symmetries so that the generative model is SE-(3) invariant; (iii) the sampling process in which the molecules are generated by the trained models; (iv) the further problems resulted from heterogenous graph with two solutions.

### 4.1   Diffusion Models

Diffusion models construct two Markov processes to learn the data distributions. The first called the forward diffusion process adds noises gradually until the noisy data's distribution reaches the prior distribution; The other called the generative denoising process, gradually removes the noise from the data sampled from the prior distribution until they are recovered to the desired data distribution. Assume there are $T$ steps in both processes, and we denote $\mathcal{M}^t = \{(s_j^t, \boldsymbol{x}_j^t, \boldsymbol{O}_j^t)\}$ as the $t$-th noisy state in the forward process, with $\mathcal{M}^T \sim \mathrm{Prior}(\mathcal{M}^T)$, and $\mathcal{M}^0 = \mathcal{M}$, where the transition distribution is denoted as $q(\cdot|\cdot)$; In the generative process, sample goes from $T$ to 0, in which the transition distribution is denoted as $p(\cdot|\cdot)$. Here we define the forward and generative processes of $s_j^t$, $\boldsymbol{x}_j^t$, and $\boldsymbol{O}_j^t$.

**Absorbing diffusion for functional group and linker types.**   Let $\boldsymbol{s}_j^t$ as the one-hot encoding of the type of a single functional group or linker. The forward process followed by D3PM [65] randomly transits $s_j^t$ into the absorbing state $K$ ($K = 23 + M_{\mathrm{fg}} + M_{\mathrm{at}}$) with

$$
\begin{aligned}
q\left(s_j^t|s_j^{t-1}\right) &= \mathrm{Multinomial}\left(\boldsymbol{s}_j^{t-1}\boldsymbol{Q}^t\right) \\
q\left(s_j^t|s_j^0\right) &= \mathrm{Multinomial}\left(\boldsymbol{s}_j^0\bar{\boldsymbol{Q}}^t\right)
\end{aligned}
\tag{1}
$$

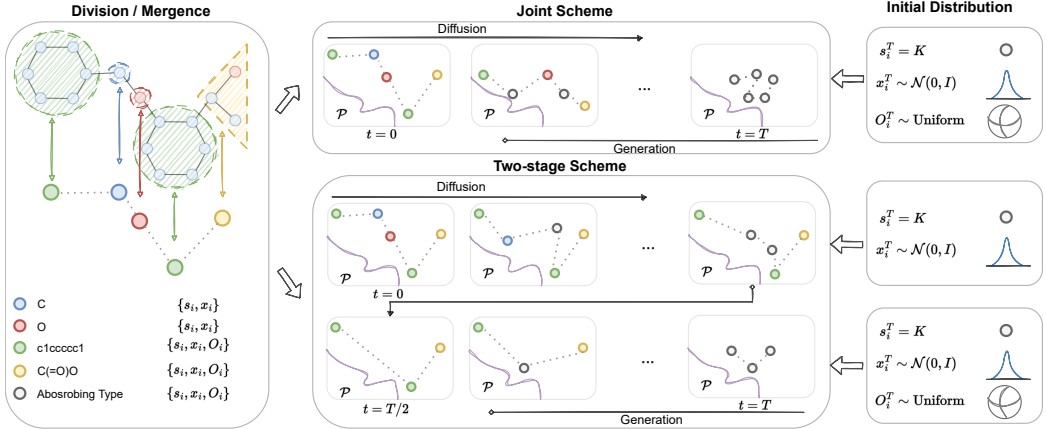

Figure 1: An illustration of the workflows of D3FG of the two schemes.

where $\bar{\boldsymbol{Q}}^t = \boldsymbol{Q}^1\boldsymbol{Q}^2\ldots\boldsymbol{Q}^t$ and $[\boldsymbol{Q}^t]_{mn} = q(s_j^t = n|s_j^{t-1} = m)$ denotes diffusion transition probabilities, with

$$[\boldsymbol{Q}^t]_{mn} = \begin{cases} 1 & \text{if} \quad m = n = K \\ 1 - \beta_{\text{type}}^t & \text{if} \quad m = n \neq K \\ \beta_{\text{type}}^t & \text{if} \quad m \neq K, n = K \end{cases} . \tag{2}$$

$\beta_{\text{type}}^t$ monotonically increases from 0 to 1, means that when $t = T$, all the type variables are absorbed into the $K$-th category. In the generative process, it first samples $N_{\text{at}}$ linkers and $N_{\text{fg}}$ functional groups whose types are all in the absorbing states, selects $(1 - \beta_{\text{type}}^t) \times 100\%$ of them respectively, and transforms their types from the absorbing states to the predicted ones by

$$p(s_j^{t-1}|\mathcal{M}^t, \mathcal{P}) = \text{Multinomial}\left(F(\mathcal{M}^t, \mathcal{P})[j]\right). \tag{3}$$

where $F$ is the type denoiser parameterized with a neural network, and $F(\cdot, \cdot)[j]$ predicts the probability of the types for the $j$-th selected functional groups or linkers. With the effectiveness of BERT-style training [66], the denoiser directly predicts $p(s_j^0|\mathcal{M}^t, \mathcal{P})$, leading to a training objective as

$$L_{\text{type}}^t = \mathbb{E}_{\mathcal{M}^t}\left[\sum_j \log p(s_j^0|\mathcal{M}^t, \mathcal{P})\right]. \tag{4}$$

**Gaussian diffusion for center atom positions.** By defining the center atom in a functional group as shown in Appendix. D, and itself as the center in a linker, the Cartesian coordinate of center nodes $\boldsymbol{x}_j$ represents its position. The forward transition distributions followed by DDPM [61] read

$$q(\boldsymbol{x}_j^t|\boldsymbol{x}_j^{t-1}) = \mathcal{N}\left(\boldsymbol{x}_j^t|\sqrt{1-\beta_{\text{pos}}^t} \cdot \boldsymbol{x}_j^{t-1}, \beta_{\text{pos}}^t\boldsymbol{I}\right);$$
$$q(\boldsymbol{x}_j^t|\boldsymbol{x}_j^0) = \mathcal{N}\left(\boldsymbol{x}_j^t|\sqrt{\bar{\alpha}_{\text{pos}}^t} \cdot \boldsymbol{x}_j^0, (1-\bar{\alpha}_{\text{pos}}^0)\boldsymbol{I}\right), \tag{5}$$

in which $\beta_{\text{pos}}^t$ increases from 0 to 1, means that the noise levels are increasing and the data's coordinate signals fade out during the forward diffusion, with $\alpha_{\text{pos}}^t = 1 - \beta_{\text{pos}}^t$, $\bar{\alpha}_{\text{pos}}^t = \alpha_{\text{pos}}^0\alpha_{\text{pos}}^1\ldots\alpha_{\text{pos}}^t$, and finally $\boldsymbol{x}_j^T \sim \mathcal{N}(\boldsymbol{0}, \boldsymbol{I})$. Note that Eq. (5) is equivalent to the Markov process of $\boldsymbol{x}_j^t = \sqrt{\bar{\alpha}_{\text{pos}}^t} \cdot \boldsymbol{x}_j^0 + \sqrt{1-\bar{\alpha}_{\text{pos}}^0} \cdot \boldsymbol{\epsilon}_j$, where $\boldsymbol{\epsilon}_j \sim \mathcal{N}(\boldsymbol{0}, \boldsymbol{I})$. Rather than predicting the mean value of the reverse transition distribution in the generative process, the position denoiser $G$ approximates the added noise $\boldsymbol{\epsilon}_j$ with the reparameterization trick as

$$p(\boldsymbol{x}_j^{t-1}|\mathcal{M}^t, \mathcal{P}) = \mathcal{N}\left(\boldsymbol{x}_j^{t-1}|\boldsymbol{\mu}_{\text{pos}}(\mathcal{M}^t, \mathcal{P}), \beta_{\text{pos}}^t\boldsymbol{I}\right);$$
$$\boldsymbol{\mu}_{\text{pos}}(\mathcal{M}^t, \mathcal{P}) = \frac{1}{\sqrt{\alpha_{\text{pos}}^t}}\left(\boldsymbol{x}_j^t - \frac{\beta_{\text{pos}}^t}{\sqrt{1-\bar{\alpha}_{\text{pos}}^t}}G(\mathcal{M}^t, \mathcal{P})[j]\right). \tag{6}$$

The training objective is thus established in a score-based way, as

$$L_{\text{pos}}^t = \mathbb{E}_{\mathcal{M}^t}\left[\sum_j \|G(\mathcal{M}^t, \mathcal{P})[j] - \boldsymbol{\epsilon}_j\|_2^2\right]. \tag{7}$$

**SO(3) diffusion for functional group orientations.** By regarding the functional groups as rigid bodies, orientations together with the center atoms' positions determine all atoms' positions. Here we represent the orientation geometry as elements in SO(3). Following [67], we use isotropic Gaussian distribution on SO(3) [68] to formulate the process, i.e. $\mathcal{IG}_{\text{so}(3)}(\cdot|\boldsymbol{\mu}_{\text{ori}}, \sigma_{\text{ori}})$, in which $\boldsymbol{\mu}_{\text{ori}}$ and $\sigma_{\text{ori}}$ are viewed as mean orientation and variance, in analogy with Gaussian distribution. The transition distribution for orientation matrices $\boldsymbol{O}_j$ reads

$$q(\boldsymbol{O}_j^t|\boldsymbol{O}_j^0) = \mathcal{IG}_{\text{so}(3)}\left(\boldsymbol{O}_j^t|\lambda_{\text{ori}}(\bar{\alpha}_{\text{ori}}^t, \boldsymbol{O}_j^0), (1 - \bar{\alpha}_{\text{ori}}^t)\right), \tag{8}$$

$\lambda_{\text{ori}}(\bar{\alpha}_{\text{ori}}^t, \boldsymbol{O}_j^0)$ is the geodesic flow from $\boldsymbol{I}$ to $\boldsymbol{O}_j^t$ by the amount $\bar{\alpha}_{\text{ori}}^t$, as $\lambda_{\text{ori}}(\bar{\alpha}_{\text{ori}}^t, \boldsymbol{O}_j^0) = \exp\left(\bar{\alpha}_{\text{ori}}^t \log(\boldsymbol{O}_j^0)\right)$, where $\exp(\cdot)$ and $\log(\cdot)$ are exponential and logarithm map on the SO(3) manifold. As $\alpha_{\text{ori}}^t \to 0$, $\lambda_{\text{ori}}(\bar{\alpha}_{\text{ori}}^t, \boldsymbol{O}_j^0) \to \boldsymbol{I}$. $\{\beta_{\text{ori}}^t\}_{t=0}^T$ is the predefined noise level schedule ranging from 0 to 1 as $t$ increases, $\alpha_{\text{ori}}^t = 1 - \beta_{\text{ori}}^t$ and $\bar{\alpha}_{\text{ori}}^t = \alpha_{\text{ori}}^0 \alpha_{\text{ori}}^1 \ldots \alpha_{\text{ori}}^t$.

In the generative process, an orientation denoiser $H$ is used to predict the mean orientation in the isotropic Gaussian distribution, which reads

$$p(\boldsymbol{O}_j^{t-1}|\mathcal{M}^t, \mathcal{P}) = \mathcal{IG}_{\text{so}(3)}\left(\boldsymbol{O}_j^{t-1}|H(\mathcal{M}^t, \mathcal{P})[j], \beta_{\text{ori}}^t\right). \tag{9}$$

We use the same loss function as in [69] to minimize the expected discrepancy measured by the inner product between the data orientation matrices and the predicted ones, which reads

$$L_{\text{ori}}^t = \mathbb{E}_{\mathcal{M}^t}\left[\sum_j \|(H(\mathcal{M}^t, \mathcal{P})[j])^\intercal \boldsymbol{O}_j^0 - \boldsymbol{I}\|_F^2\right]. \tag{10}$$

### 4.2 Parametrization of Denoisers with Neural Networks

**Amino acid context encoding.** In order to decrease the computational complexity, we denote the protein context at amino-acid levels. Besides the amino acid types, $C_\alpha$ atom coordinate and the orientation, each atom's coordinates in the local system and three torsion angles including angles around 'N-$C_\alpha$' bond, '$C_\alpha$-C' bond and 'C-N' bond, are also used as intra-amino-acid features, which are concatenated and embedded by an MLP to create the intra-amino-acid embedding vector $\boldsymbol{e}_i$. For inter-amino-acid features, the pair of amino acid types, sequential relationships (if the two amino acids are adjacent in the protein sequence), pairwise distances between $C_\alpha$ and inter-residue dihedrals are all embedded as inter-amino-acid embedding vector $\boldsymbol{z}_{i,j}$ with $i, j \in \mathcal{I}_{\text{aa}}$. Note that these embedding vectors are all translational and rotational invariant (Appendix. C).

**Denoisers with equivariance.** In the setting of generative models, the learned distribution $p(\mathcal{M}|\mathcal{P})$ should be equivariant to translation and rotation, such that $p(\mathbf{T}_g(\mathcal{M})|\mathbf{T}_g(\mathcal{P})) = p(\mathcal{M}|\mathcal{P})$ for any $g \in \text{SE}(3)$, where $\mathbf{T}_g$ is the corresponding roto-translational transformations, and $\mathbf{T}_g(\mathcal{M})$ means each atom in the molecule is rotated and translated by $\mathbf{T}_g$. In the setting of diffusion models, the following proposition indicates the translational and rotational equivariance of each denoisers.

**Proposition 1.** Let $p(\boldsymbol{x}^T)$, $p(\boldsymbol{O}^T)$, and $p(s^T)$ be SE(3)-invariant distribution, and the transition distributions $p(\boldsymbol{x}^{t-1}|\mathcal{M}^t, \mathcal{P})$ be SE(3)-equivariant, $p(\boldsymbol{O}^{t-1}|\mathcal{M}^t, \mathcal{P})$ be T(3)-invariant and SO(3)-equivariant, and $p(s^{t-1}|\mathcal{M}^t, \mathcal{P})$ be SE(3)-invariant, then the density $p(\mathcal{M}|\mathcal{P})$ modeled by the reverse Markov Chains in the generative process of diffusion models is SE(3)-equivariant.

According to the **Proposition 1**, while the denoisers for functional group and linker types are roto-translational invariant, the denoiser for positions should be roto-translational equivariant, and it for orientations should be translational invariant and rotational equivariant. Therefore, we employ our denoiser's network structures with IPA [3] by harnessing the expressivity of TRANSFORMER [70] and roto-translational equivariance of LoCS [71]. Denote the binding system as a graph, in which the nodes are composed of amino acids, functional groups, and linkers, and the edges are established with

**Algorithm 1** Joint Generation for Molecules using D3FG
___
**Input:** Zero-centered protein $\{s_i, \boldsymbol{x}_i, \boldsymbol{O}_i\}_{i \in \mathcal{I}_{aa}}$, and graph denoiser $F, G, H$, and node number sampler $\phi_{fg}, \phi_{at}$.
Sample $N_{fg} \sim \phi_{fg}, N_{at} \sim \phi_{at}$, leading to the index set $\mathcal{I}_{fg}$ and $\mathcal{I}_{at}$.
Sample initial states of functional groups, $\{s_j^T, \boldsymbol{x}_j^T, \boldsymbol{O}_j^T\}_{j \in \mathcal{I}_{fg} \cup \mathcal{I}_{at}}$, where $s_j^t = K$, $\boldsymbol{x}_j^T \sim \mathcal{N}(\boldsymbol{0}, \boldsymbol{I})$,
$\boldsymbol{O}_j^T \sim \text{Uniform}_{SO(3)}$ for $j \in \mathcal{I}_{fg}$ else $\boldsymbol{O}_j^T = \boldsymbol{I}$.
**for** $t$ in $T-1, T-2, \dots, 1$ **do**
    Sample $\{s_j^{t-1}, \boldsymbol{x}_j^{t-1}, \boldsymbol{O}_j^{t-1}\}_{j \in \mathcal{I}_{fg} \cup \mathcal{I}_{at}}$ as Eq. (3), (6) and (9) and update $\mathcal{M}^{t-1}$.
**end for**
**Output:** $\mathcal{M} = \{s_j^0, \boldsymbol{x}_j^0, \boldsymbol{O}_j^0\}_{j \in \mathcal{I}_{fg} \cup \mathcal{I}_{at}}$
___

K-nearest neighbor. Let $\{\boldsymbol{h}_i : i \in \mathcal{I}_{aa} \cup \mathcal{I}_{fg} \cup \mathcal{I}_{at}\}$ be the node embedding which is SE(3)-invariant, $\{\boldsymbol{e}_i : i \in \mathcal{I}_{aa} \cup \mathcal{I}_{fg} \cup \mathcal{I}_{at}\}$ and $\{\boldsymbol{z}_{i,j} : i, j \in \mathcal{I}_{aa} \cup \mathcal{I}_{fg} \cup \mathcal{I}_{at}\}$ be the previously defined intra- and inter-amino acid embedding, with $\boldsymbol{e}_j = \boldsymbol{0}$ and $\boldsymbol{z}_{i,j} = \boldsymbol{0}$ if $i \notin \mathcal{I}_{aa}$ or $j \notin \mathcal{I}_{aa}$. The attention mechanism in IPA updates the embedding of node $i$ as

$$\boldsymbol{h}_i' = \sum_{j \in \mathcal{N}(i)} \frac{\exp\left((\boldsymbol{W}_q \boldsymbol{e}_j')^\intercal (\boldsymbol{W}_k \boldsymbol{e}_i') + \boldsymbol{z}_{i,j}\right)(\boldsymbol{W}_v \boldsymbol{e}_j')}{\sum_{j \in \mathcal{N}(i)} \exp\left((\boldsymbol{W}_q \boldsymbol{e}_j')^\intercal (\boldsymbol{W}_k \boldsymbol{e}_i') + \boldsymbol{z}_{i,j}\right)} \tag{11}$$

where $\boldsymbol{e}_i' = \boldsymbol{h}_i + \boldsymbol{e}_i$, $\boldsymbol{h}_i'$ is the updated node embedding, $\boldsymbol{W}_q, \boldsymbol{W}_k, \boldsymbol{W}_v$ are learnable parameters, and $\mathcal{N}(i)$ is neighborhood of node $i$ obtained by the edges. Because $\boldsymbol{h}_i, \boldsymbol{e}_i$, and $\boldsymbol{z}_{i,j}$ are all SE(3)-invariant, $\boldsymbol{h}_i'$ is also invariant. Three heads parameterized with MLP are stacked after several layers of Transformers update the node embedding, denoted by $\text{MLP}_F(\cdot)$, $\text{MLP}_G(\cdot)$, and $\text{MLP}_H(\cdot)$ is used for updating $s^{t-1}$, $\boldsymbol{x}^{t-1}$ and $\boldsymbol{O}^{t-1}$. The LoCS updates the parameters in Eq. (3), (6) and (9) by

$$\begin{aligned} F\left(\mathcal{M}^t, \mathcal{P}\right)[j] &= \text{MLP}_F(\boldsymbol{h}_j') \\ G\left(\mathcal{M}^t, \mathcal{P}\right)[j] &= \text{MLP}_G(\boldsymbol{h}_j')\boldsymbol{O}_j^t \\ H\left(\mathcal{M}^t, \mathcal{P}\right)[j] &= \exp\left(\text{MLP}_H(\boldsymbol{h}_j')\right)\boldsymbol{O}_j^t \end{aligned} \tag{12}$$

The output of $G$ means predicting the coordinate deviations in the local coordinate systems and then projecting it into the global one; In $H$, $\text{MLP}_H$ first predicts a vector in Lie group $\mathfrak{so}(3)$ and the exponential map on SO(3) converts it into a rotation matrix. The updating process ensures the equivariance and the invariance of the transition distributions in **Proposition 1**, which is proved in [71] and [69]. However, since $\boldsymbol{O}_i^t = \boldsymbol{I}$ for any $i \in \mathcal{I}_{at}$, the equivariance of $G$ on linkers will not be preserved, so we instead use EGNN [44] to get the output of $G$ (Appendix. C).

### 4.3 Sampling Process

In sampling, we first use two prior distributions as the empirical distributions $\phi_{at}$ and $\phi_{fg}$ drawn from the training set to sample the linker $N_{at}$ and functional group number $N_{fg}$. As $p(\boldsymbol{x}^T), p(\boldsymbol{O}^T)$ and $p(s^T)$ should be SE(3)-invariant, $p(s^T) = \mathbb{1}_{\{K\}}(s^T)$, $p(\boldsymbol{x}^T) = \mathcal{N}(\boldsymbol{x}^T|\boldsymbol{0}, \boldsymbol{I})$ and $p(\boldsymbol{O}^T) = \text{Uniform}_{SO(3)}(\boldsymbol{O}^T)$ satisfy the conditions, where $\mathbb{1}.(\cdot)$ is the indicator function and $\text{Uniform}_{SO(3)}(\cdot)$ is the uniform distribution on SO(3). And the iteratively generative process of $p(s^{t-1}|\mathcal{M}^t, \mathcal{P})$, $p(\boldsymbol{x}^{t-1}|\mathcal{M}^t, \mathcal{P})$ and $p(\boldsymbol{O}^{t-1}|\mathcal{M}^t, \mathcal{P})$ are given in Eq. (3), (6) and (9). The detailed sampling algorithm is given in Algorithm. 1.

### 4.4 Heterogeneous graph: Joint or Two-stage?

Amino acids and functional groups are both fragments composed of atoms in proteins and molecules, regarded as rigid bodies, while the linkers are single atoms, regarded as mass points. Therefore, in the binding graph, nodes are of different levels and connections are of different kinds, thus leading the graph to be heterogeneous. For this reason, we propose two generative schemes: joint generation scheme and two-stage generation scheme as shown in Figure. 1.

In joint scheme, we regard amino acids, functional groups, and linkers at the same level, and use one single neural network to predict the three variables and update them. In detail, this scheme directly

models $p(\mathcal{M}|\mathcal{P}) = p(\{s_i, \boldsymbol{x}_i, \boldsymbol{O}_i\}_{i \in \mathcal{I}_{\text{fg}} \cup \mathcal{I}_{\text{at}}}|\mathcal{P})$, where the parameterized transition distribution is $p(\{s_i^{t-1}, \boldsymbol{x}_i^{t-1}, \boldsymbol{O}_i^{t-1}\}_{i \in \mathcal{I}_{\text{fg}} \cup \mathcal{I}_{\text{at}}}|\mathcal{M}^t, \mathcal{P})$. Note that $\boldsymbol{O}_i^t = \boldsymbol{I}$ for any $t$ if $i \in \mathcal{I}_{\text{at}}$.

In two-stage scheme, we regard amino acids and functional groups at the fragment level, and linkers at the atom level, and use two different neural networks to parameterize the transition distribution. In the first stage, the functional groups are generated, and then single atoms as linkers will be generated to connect the generated functional groups as a full molecule. The two-stage generative scheme is similar to CADD, which first determines the pharmacophores towards the target protein, fits functional groups with high activity into them, and then searches for the possible molecules with these functional groups. In specific, the generative process reads $p(\mathcal{M}|\mathcal{P}) = p(\{s_j, \boldsymbol{x}_j\}_{j \in \mathcal{I}_{\text{at}}}|\{s_i, \boldsymbol{x}_i, \boldsymbol{O}_i\}_{i \in \mathcal{I}_{\text{fg}}}, \mathcal{P})p(\{s_i, \boldsymbol{x}_i, \boldsymbol{O}_i\}_{i \in \mathcal{I}_{\text{fg}}}|\mathcal{P})$. The transition distribution of the first stage $p(\{s_i^{t-1}, \boldsymbol{x}_i^{t-1}, \boldsymbol{O}_i^{t-1}\}_{i \in \mathcal{I}_{\text{fg}}}|\{s_i^t, \boldsymbol{x}_i^t, \boldsymbol{O}_i^t\}_{i \in \mathcal{I}_{\text{fg}}}, \mathcal{P})$ is parameterized by one neural network; In the second stage, the generated $\{s_i, \boldsymbol{x}_i, \boldsymbol{O}_i\}_{i \in \mathcal{I}_{\text{fg}}}$ is used as context, so that the other neural network models $p(\{s_i^{t-1}, \boldsymbol{x}_i^{t-1}\}_{i \in \mathcal{I}_{\text{at}}}|\{s_i^t, \boldsymbol{x}_i^t\}_{i \in \mathcal{I}_{\text{at}}}, \{s_i, \boldsymbol{x}_i, \boldsymbol{O}_i\}_{i \in \mathcal{I}_{\text{fg}}}, \mathcal{P})$.

## 5 Experiments

### 5.1 Data Processing.

In the experiments, we use CrossDocked2020[32] for evaluation. In the prevailing works, they focus on generating molecules at the atom level, differing from our functional-group-based generation, so our first target is to divide the molecules into functional groups and linkers. We use EFGs[72] to segment molecules. We select the 25 functional groups that appear most frequently with stable structures, which are partly shown in Figure. 2 For some functional groups, chirality exists in their structures, and we treat them as two functional groups. As a result, we finally build up a dataset as a corpus including 27 functional groups (two of the 25 have chirality) for Crossdocked2020, with their intra-structures fixed as rigid bodies, and assure that most molecules can be decomposed into the substructures in our corpus datasets. Details are given in Appendix. D. For linkers, we choose $\{\text{B}, \text{C}, \text{N}, \text{O}, \text{F}, \text{P}, \text{S}, \text{Cl}, \text{Br}, \text{I}\}$ as representative heavy atoms. After the processing, we obtain $M_{\text{fg}} = 27$ and $M_{\text{at}} = 10$ in our experiments. Besides, by the fragment-linker designation of a binding graph, the node number is reduced to 53.62 on average in GNN's message-passing.

### 5.2 Pocket-Specific Molecule Generation

**Dataset.** The datasets for training and evaluation are split according to POCKET2MOL [9] and TARGETDIFF [13]. 22.5 million docked protein binding complexes with low RMSD ( $< 1\text{Å}$ ) and sequence identity less than $30\%$ are selected, leading to 100,000 pairs of pocket-ligand complexes, with 100 novel complexes as references for evaluation.

**Setup.** For performance comparison, our methods are compared with baselines including LIGAN [60], 3DSBDD [12], GRAPHBP [10], POCKET2MOL [54], DIFFSBDD [14] and TARGETDIFF [13]. LIGAN as a 3D CNN-based method generates atoms on regular grids, with VAE [73] as its generative model. 3DSBDD, GRAPHBP and POCKET2MOL are all GNN-based, generating atoms in an auto-regressive way. DIFFSBDD and TARGETDIFF are two diffusion-based methods that generate molecules at the full atom level, with equivariant GNNs as the denoisers. The two schemes of generation lead to two variants of our method, written as D3FG(Joint) and D3FG(Stage). In some parts, we choose POCKET2MOL as the representative autoregressive methods, and DIFFSBDD and TARGETDIFF as the benchmarks employing diffusion models, because these three baselines are the

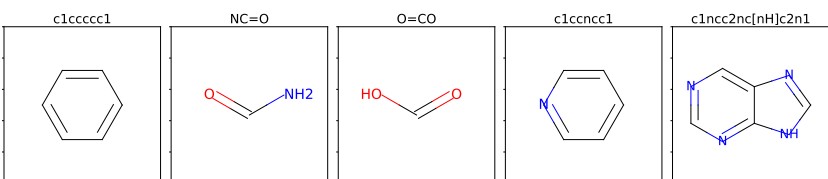

Figure 2: Five of twenty-five functional groups with stable structures that occur most frequently in Crossdocked2020 and are used in D3FG.

Table 1: 'Ratio' of the top ten functional groups with the highest frequency in Crossdocked2020. 'Ref.' is calculated in the training set. MAE is obtained between rows of 'Ref.' and different methods' 'Ratio' and JSD is calculated by 'Freq', which is in detail given in Appendix. E 'Ratio' in **bold** means it is the closest to 'Ref.', and MAE/JSD in **bold** means it is the lowest.

| Functional Group | Ref. | POCKET2MOL | TARGETDIFF | DIFFSBDD | D3FG (Joint) | D3FG (Stage) |
|---|---|---|---|---|---|---|
| c1ccccc1 | 0.712 | 0.583 | 0.293 | 0.131 | 0.548 | **0.608** |
| NC=O | 0.266 | 0.089 | 0.149 | 0.010 | 0.120 | **0.159** |
| O=CO | 0.216 | 0.200 | 0.320 | 0.025 | **0.226** | 0.127 |
| c1ccncc1 | 0.082 | 0.086 | 0.052 | 0.001 | 0.040 | **0.078** |
| c1ncc2nc[nH]c2n1 | 0.061 | 0.001 | 0.000 | 0.000 | 0.002 | **0.030** |
| NS(=O)=O | 0.055 | 0.000 | 0.000 | **0.001** | **0.001** | **0.001** |
| O=P(O)(O)O | 0.040 | 0.004 | **0.020** | 0.000 | 0.011 | 0.015 |
| OCO | 0.034 | **0.024** | 0.097 | 0.001 | 0.067 | 0.075 |
| c1cncnc1 | 0.032 | 0.010 | **0.015** | 0.000 | 0.003 | 0.013 |
| c1cn[nH]c1 | 0.029 | **0.013** | 0.002 | 0.000 | 0.004 | 0.006 |
| MAE (↓) | - | 0.030 | 0.045 | 0.071 | 0.024 | **0.020** |
| JSD(↓) | - | 0.248 | 0.301 | 0.553 | 0.223 | **0.201** |

Table 2: Jensen-Shannon divergence between the distributions of bond distance for reference vs. generated molecules. The smaller, the better. Value in **bold** is the lowest.

| Bond | LIGAN | 3DSBDD | POCKET2MOL | TARGETDIFF | DIFFSBDD | D3FG (Joint) | D3FG (Stage) |
|---|---|---|---|---|---|---|---|
| C-C | 0.599 | 0.424 | 0.416 | 0.346 | 0.385 | 0.339 | **0.281** |
| C=C | 0.665 | 0.545 | 0.516 | 0.503 | 0.565 | 0.485 | **0.469** |
| C-N | 0.631 | 0.424 | 0.401 | **0.299** | 0.421 | 0.307 | 0.313 |
| C=N | 0.742 | 0.671 | 0.628 | 0.547 | 0.569 | 0.530 | **0.523** |
| C-O | 0.656 | 0.547 | 0.445 | 0.408 | 0.413 | 0.412 | **0.406** |
| C=O | 0.662 | 0.638 | 0.532 | **0.467** | 0.541 | 0.490 | 0.488 |
| c:c | 0.494 | 0.410 | 0.398 | **0.264** | 0.339 | 0.327 | 0.302 |
| c:n | 0.634 | 0.443 | 0.457 | **0.228** | 0.260 | 0.246 | 0.237 |

latest works that show good empirical performance. We sample 100 valid molecules for each pocket for the baselines and D3FG, leading to 10,000 pairs of complexes. After the molecules are generated by the model, `Openbabel` [74] is used to construct chemical bonds between atoms, and Universal Force Field (UFF) minimization [75] is used for refinement. The following evaluations are based on the samples. Figure. 4 gives an example of generated molecules by different methods.

**Structure analysis.** We first analyze the **functional groups** of generated molecules by different methods. We show the 'Ratio' and 'Freq' of the included functional groups partly, where 'Ratio' means how many specified functional groups are in each molecule on average, and 'Freq' means the statistical frequency of occurrence of the specified functional group in the generated molecules. Mean absolute error (MAE) as overall metrics is calculated according to reference 'Ratio' and generated 'Ratio'; Jensen-Shannon divergence (JSD) is calculated according to reference 'Freq' and generated 'Freq'. The smaller MAE and JSD are, the better performance the method achieves. Table. 1 shows the metrics of different methods' generated molecules, demonstrating D3FG's superiority in generating molecules with realistic drug structures since distributions of the functional groups in generated molecules are more similar to the real drug molecules'. Secondly, we analyze the **atom type**, **bond distance**, **bond angle**, **dihedral** and **atom type** distribution. JSDs are calculated between the distributions of bond distance for reference vs. molecules. '-', '=',':' represents single, double, and aromatic bonds, respectively. Besides, we report MAE on reference 'Ratio' vs. generated 'Ratio' and JSD on reference 'Freq' vs. generated 'Freq' based on atom type distribution. Table. 2, 3 and Figure. 3 gives details atom type analysis. For other geometries, please refer to Appendix. E. We found that D3FG(Stage) outperforms D3FG(Joint) in generating more realistic molecules, and has competitive performance with TARGETDIFF.

**Binding affinity.** We secondly make a comparison of Binding Affinity. Following 3DSBDD and LIGAN, we employed two evaluation metrics including Vina docking score [76] and Gnina docking score [77]. Vina docking [78] as a classical docking tool, gives a lower score as Vina energy if the binding affinity of the molecule is better, while Gnina docking as a deep-learning-based docking tool, gives it a higher score. ΔAffinity measures the percentage of generated molecules that have better predicted binding affinity than the corresponding reference molecules. Table. 4 shows that our D3FG

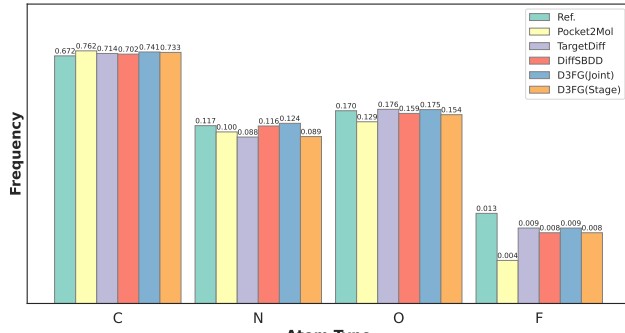

Figure 3: Atom type distribution and metrics.

| Method | MAE(↓) | JSD(↓) |
|---|---|---|
| POCKET2MOL | 0.573 | 0.098 |
| TARGETDIFF | 0.471 | 0.059 |
| DIFFSBDD | 0.627 | **0.054** |
| D3FG(Joint) | 0.528 | 0.075 |
| D3FG(Stage) | **0.294** | 0.056 |

Table 3: MAE and JSD of different methods' atom type number and distribution vs. the reference. Values in **bold** are the lowest.

Table 4: Evaluation of Binding affinity and other chemical drug properties for baselines and variants of D3FG. ↓ means the smaller the value, the better the performance, and ↑ means the opposite. Values in **bold** are the top-2 best metrics.

| | Vina Score (↓) | Vina ΔAffinity (↑) | Gnina Score (↑) | Gnina ΔAffinity(↑) | QED (↑) | SA (↑) | LogP | Lipinski (↑) |
|---|---|---|---|---|---|---|---|---|
| Ref. (Test) | -7.06 | - | 5.37 | - | 0.471 | 0.725 | 0.818 | 4.247 |
| LIGAN | -6.17 | 21.24% | 4.29 | 21.68% | 0.382 | 0.584 | -0.138 | 4.046 |
| GRAPHBP | -6.36 | 27.41% | 4.52 | 26.54% | 0.437 | 0.502 | 3.024 | 4.448 |
| 3DSBDD | -6.12 | 20.73% | 4.48 | 19.22% | 0.426 | 0.625 | 0.266 | 4.735 |
| POCKET2MOL | -6.92 | 45.86% | 5.34 | 40.68% | **0.543** | 0.746 | 1.584 | 4.904 |
| TARGETDIFF | **-7.11** | **49.52%** | **5.41** | 42.40% | 0.474 | 0.581 | 1.402 | 4.487 |
| DIFFSBDD | -6.37 | 31.32% | 4.63 | 27.96% | 0.494 | 0.343 | -0.157 | 4.895 |
| D3FG (Joint) | -6.89 | 37.32% | 5.30 | 33.45% | **0.507** | **0.832** | 2.796 | **4.943** |
| D3FG (Stage) | **-6.96** | **45.88%** | **5.43** | **43.36%** | 0.501 | **0.840** | 2.821 | **4.965** |
| D3FG (EHot) | -7.19 | 51.78% | 5.51 | 56.53% | 0.482 | 0.731 | 0.814 | 4.330 |
| D3FG (ECold) | -7.02 | 44.03% | 5.16 | 32.69% | 0.476 | 0.707 | 0.820 | 4.228 |

of the two-stage scheme achieves competitive affinity scores, comparable to TARGETDIFF and much better than D3FG of the joint scheme.

**Drug property.** Moreover, chemical properties are evaluated with RdKit [79], including QED [80] (quantitative estimation of drug-likeness), SA [81] (synthetic accessibility score), LogP [82] (the octanol-water partition coefficient, which should be between $-0.4$ and $5.6$ for good drug candidates), and Lipinski [83, 84] (number of rules the drug follow the Lipinski's rule of five). QED, SA, and Lipinski are three metrics with preferences for atom numbers, demonstrated in DIFFBP[15]. Table. 4 demonstrates that two variants of D3FG achieve overall best performance in these metrics.

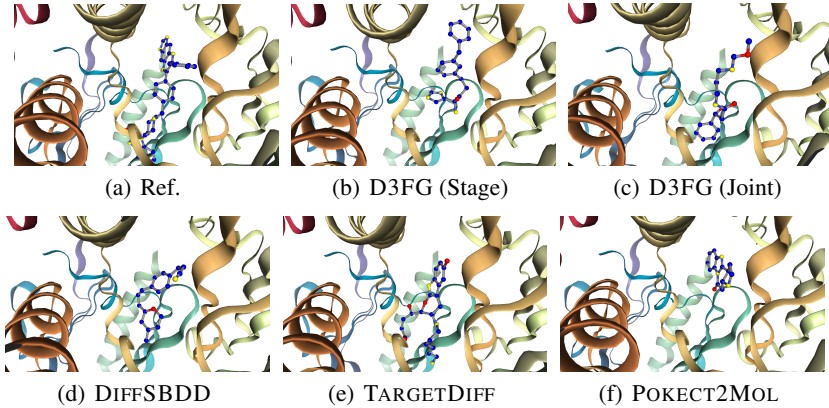

(a) Ref.      (b) D3FG (Stage)      (c) D3FG (Joint)

(d) DIFFSBDD      (e) TARGETDIFF      (f) POKECT2MOL

Figure 4: Generated molecules by different methods on pocket `3o96_A_rec`. The diffusion-based methods generated molecules more similar to the reference, appearing to be 'vertical'.

### 5.3 Pocket-Specific Molecule Elaboration

**Introduction.**    Molecule elaboration as a sub-task of molecule optimization, aims to elaborate a fragment of existing molecules amenable to chemical modification for improving binding affinity. To fulfill it, we here first select a functional group in a ligand that contributes to binding affinity and remove it to obtain the editable fragments. Then, we attempt to use D3FG to generate the new type of functional groups with its structures, for modifying the fragments and thus building up a new molecule with higher binding affinity to the target protein.

**Dataset.**    The selection of a functional group for replacement is the first problem. Pharmacophoric information is extracted by calculating the fragment hotspot map (FHM) [33, 34] of the target protein. In specific, FHM describes regions of the binding pocket that are likely to make a positive contribution to binding affinity. Then, by placing the molecules into the pockets, we can thus obtain each functional group's hotspot scores, according to the binding complexes. The higher hotspot score a functional group reaches, the more contributions it makes to the binding affinity. Functional groups' hotspot scores of each ligand are calculated based on 100,000 pairs of pocket-ligand complexes in Crossdocked2020, and the selection of functional groups can be based on their scores. Finally, we established our new datasets for molecule elaboration based on FHM.

**Binding affinity and drug property.**    We here consider two schemes for molecule elaboration, the first is to remove the functional groups with the highest scores, and elaborate the remaining fragments by replacing them with the functional group generated by D3FG. We write this as D3FG(EHot). The second D3FG(ECold), on the other hand, replaces the functional groups of the lowest scores). We report the D3FG's elaborated molecules by the two schemes in Table. 4. It shows that D3FG(EHot) tends to generate more molecules with higher affinity, while molecules elaborated by D3FG(ECold) show tiny differences in binding affinity from the raw references. Besides, on other chemical properties, the elaborated molecules are very close to the raw references, since the differences lie only in one single functional group, and the major molecular skeletons are almost identical.

## 6 Conclusion and Future Work

In this paper, a functional-group-based diffusion model called D3FG is proposed to generate molecules in 3D with protein structures as its context. Joint and two-stage generation schemes lead to two variants of D3FG. The molecules generated by the two-stage generation scheme show more realistic structures, competitive binding performance, and better drug properties. Besides, in molecule elaboration, D3FG can also generate molecules with good binding affinity. However, limitation still exists. First, the functional group datasets are still small, which will be enlarged in the future. Second, the binding affinities of generated molecules still remain to be improved, since other diffusion models even show better binding performance.

## Acknowledgements

This work was supported by National Key R&D Program of China (No. 2022ZD0115100), National Natural Science Foundation of China Project (No. U21A20427), and Project (No. WU2022A009) from the Center of Synthetic Biology and Integrated Bioengineering of Westlake University.

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

# A    Proof of Proposition 1.

First, we here denote all atom's positions in the molecules as $\boldsymbol{X}_{\mathrm{M}} \in \mathbb{R}^{N_{\mathrm{a}} \times 3}$, and in the protein as $\boldsymbol{X}_{\mathrm{M}} \in \mathbb{R}^{N_{\mathrm{aa}} \times 3}$, and linker and functional group type as $\boldsymbol{S}_{\mathrm{M}} \in \mathbb{R}^{(N_{\mathrm{aa}} + N_{\mathrm{fg}}) \times (22 + M_{\mathrm{fg}} + M_{\mathrm{at}})}$. Note that one functional group may contain several atoms so that $N_{\mathrm{aa}} + N_{\mathrm{fg}} < N_{\mathrm{a}}$.

SE(3) group as a roto-translation group in $\mathbb{R}^3$, can be divided into two groups: SO(3) as the rotation group and T(3) as the translation group. For $\boldsymbol{x} \in \mathbb{R}^3$, and $g = r + v$ with $g \in \mathrm{SE}(3), r \in \mathrm{SO}(3), v \in \mathrm{T}(3)$, $T_g(\boldsymbol{x}) = T_{r+v}(\boldsymbol{x}) = T_v \circ T_r(\boldsymbol{x})$

**Lemma A1.** If the The equivariance and invariance of the distribution in the reverse diffusion process are listed as

$$p\left(\mathrm{T}_g(\boldsymbol{X}_{\mathrm{M}}^T), \boldsymbol{S}_{\mathrm{M}}^T | \mathrm{T}_g(\boldsymbol{X}_{\mathrm{P}}), \boldsymbol{S}_{\mathrm{P}}\right) = p\left(\boldsymbol{X}_{\mathrm{M}}^T, \boldsymbol{S}_{\mathrm{M}}^T\right) = p\left(\boldsymbol{X}_{\mathrm{M}}^T, \boldsymbol{S}_{\mathrm{M}}^T | \boldsymbol{X}_{\mathrm{P}}, \boldsymbol{S}_{\mathrm{P}}\right)$$
$$p\left(\mathrm{T}_g(\boldsymbol{X}_{\mathrm{M}}^{t-1}) | \mathrm{T}_g(\boldsymbol{X}_{\mathrm{M}}^t), \boldsymbol{S}_{\mathrm{M}}^t, \mathrm{T}_g(\boldsymbol{X}_{\mathrm{P}}), \boldsymbol{S}_{\mathrm{P}}\right) = p\left(\boldsymbol{X}_{\mathrm{M}}^{t-1} | \boldsymbol{X}_{\mathrm{M}}^t, \boldsymbol{S}_{\mathrm{M}}^t, \boldsymbol{X}_{\mathrm{P}}, \boldsymbol{S}_{\mathrm{P}}\right) \quad (13)$$
$$p\left(\boldsymbol{S}_{\mathrm{M}}^{t-1} | \mathrm{T}_g(\boldsymbol{X}_{\mathrm{M}}^t), \boldsymbol{S}_{\mathrm{M}}^t, \mathrm{T}_g(\boldsymbol{X}_{\mathrm{P}}), \boldsymbol{S}_{\mathrm{P}}\right) = p\left(\boldsymbol{S}_{\mathrm{M}}^{t-1} | \boldsymbol{X}_{\mathrm{M}}^t, \boldsymbol{S}_{\mathrm{M}}^t, \boldsymbol{X}_{\mathrm{P}}, \boldsymbol{S}_{\mathrm{P}}\right),$$

Then $p\left(\boldsymbol{X}_{\mathrm{M}}^T, \boldsymbol{S}_{\mathrm{M}}^T | \boldsymbol{X}_{\mathrm{P}}, \boldsymbol{S}_{\mathrm{P}}\right)$ is SE(3) invariant.

*Proof.* Since $\mathbf{T}_g(\mathcal{M}) = \{\boldsymbol{S}, \mathrm{T}_g(\boldsymbol{X}_{\mathrm{M}})\}$, we can write the joint generative process as

$$p\left(\mathrm{T}_g(\boldsymbol{X}_{\mathrm{M}}), \boldsymbol{S}_{\mathrm{M}} | \mathrm{T}_g(\boldsymbol{X}_{\mathrm{P}}), \boldsymbol{S}_{\mathrm{P}}\right)$$
$$= \int p\left(\mathrm{T}_g(\boldsymbol{X}_{\mathrm{M}}^T), \boldsymbol{S}_{\mathrm{M}}^T | \mathrm{T}_g(\boldsymbol{X}_{\mathrm{P}}), \boldsymbol{S}_{\mathrm{P}}\right) \prod_{t=0}^{T-1} p\left(\mathrm{T}_g(\boldsymbol{X}_{\mathrm{M}}^{t-1}), \boldsymbol{S}_{\mathrm{M}}^{t-1} | \mathrm{T}_g(\boldsymbol{X}_{\mathrm{M}}^t), \boldsymbol{S}_{\mathrm{M}}^t, \mathrm{T}_g(\boldsymbol{X}_{\mathrm{P}}), \boldsymbol{S}_{\mathrm{P}}\right) d\mathcal{M}^{0:T-1}$$
$$= \int p\left(\boldsymbol{X}_{\mathrm{M}}^T, \boldsymbol{S}_{\mathrm{M}}^T | \boldsymbol{X}_{\mathrm{P}}, \boldsymbol{S}_{\mathrm{P}}\right) \prod_{t=0}^{T-1} p\left(\mathrm{T}_g(\boldsymbol{X}_{\mathrm{M}}^{t-1}), \boldsymbol{S}_{\mathrm{M}}^{t-1} | \mathrm{T}_g(\boldsymbol{X}_{\mathrm{M}}^t), \boldsymbol{S}_{\mathrm{M}}^t, \mathrm{T}_g(\boldsymbol{X}_{\mathrm{P}}), \boldsymbol{S}_{\mathrm{P}}\right) d\mathcal{M}^{0:T-1}$$
$$= \int p\left(\boldsymbol{X}_{\mathrm{M}}^T, \boldsymbol{S}_{\mathrm{M}}^T | \boldsymbol{X}_{\mathrm{P}}, \boldsymbol{S}_{\mathrm{P}}\right) \prod_{t=0}^{T-1} p\left(\mathrm{T}_g(\boldsymbol{X}_{\mathrm{M}}^{t-1}) | \mathrm{T}_g(\mathcal{M}^t, \mathcal{P})\right) p\left(\boldsymbol{S}_{\mathrm{M}}^{t-1} | \mathrm{T}_g(\mathcal{M}^t, \mathcal{P})\right) d\mathcal{M}^{0:T-1}$$
$$= \int p\left(\boldsymbol{X}_{\mathrm{M}}^T, \boldsymbol{S}_{\mathrm{M}}^T | \boldsymbol{X}_{\mathrm{P}}, \boldsymbol{S}_{\mathrm{P}}\right) \prod_{t=0}^{T-1} p\left(\boldsymbol{X}_{\mathrm{M}}^{t-1} | \mathcal{M}^t, \mathcal{P}\right) p\left(\boldsymbol{S}_{\mathrm{M}}^{t-1} | \mathcal{M}^t, \mathcal{P}\right) d\mathcal{M}^{0:T-1}$$
$$= \int p\left(\boldsymbol{X}_{\mathrm{M}}^T, \boldsymbol{S}_{\mathrm{M}}^T | \boldsymbol{X}_{\mathrm{P}}, \boldsymbol{S}_{\mathrm{P}}\right) \prod_{t=0}^{T-1} p\left(\boldsymbol{X}_{\mathrm{M}}^{t-1}, \boldsymbol{S}_{\mathrm{M}}^{t-1} | \boldsymbol{X}_{\mathrm{M}}^t, \boldsymbol{S}_{\mathrm{M}}^t, \boldsymbol{X}_{\mathrm{P}}, \boldsymbol{S}_{\mathrm{P}}\right) d\mathcal{M}^{0:T-1}$$
$$= p\left(\boldsymbol{X}_{\mathrm{M}}, \boldsymbol{S}_{\mathrm{M}} | \boldsymbol{X}_{\mathrm{P}}, \boldsymbol{S}_{\mathrm{P}}\right)$$

$$(14)$$

Then, let's consider a single atom's position. We here denote each atom's $\boldsymbol{x}_{\mathrm{M}}^t$ as

$$\boldsymbol{x}_{\mathrm{M}}^t = \boldsymbol{x}_{\mathrm{C}}^t + \boldsymbol{x}_{\mathrm{R}}^t \boldsymbol{O}_c^t \quad (15)$$

where $\boldsymbol{x}_{\mathrm{C}}^t$ is the defined center atom's position in the functional group, $\boldsymbol{x}_{\mathrm{R}}^t$ is the relative position of the atom in the local coordinate system centering at $\boldsymbol{x}_{\mathrm{C}}^t$, $\boldsymbol{O}_{\mathrm{C}}^t$ is the rotation matrices of the local coordinate system with respect to the global system. Moreover, because the functional group is regarded as rigid bodies, $\boldsymbol{x}_{\mathrm{R}}^t = \boldsymbol{x}_{\mathrm{R}}$ is constant. To be specific, if $\boldsymbol{x}_{\mathrm{M}}^t$ refers to linker's position, $\boldsymbol{O}_{\mathrm{C}}^t = \boldsymbol{0}$.

**Proposition A2.** If each atom's relative positions in the local coordinate systems are fixed, and $p(\boldsymbol{x}_{\mathrm{C}}^{t-1} | \mathcal{M}^t, \mathcal{P})$ is SE(3)-equivariant and $p(\boldsymbol{O}_{\mathrm{C}}^{t-1} | \mathcal{M}^t, \mathcal{P})$ is SO(3)-equivariant and T(3)-invariant, such that $p\left(\mathrm{T}_g(\boldsymbol{x}_{\mathrm{C}}^{t-1}) | \mathrm{T}_g(\mathcal{M}^t, \mathcal{P})\right) = p\left(\boldsymbol{x}_{\mathrm{C}}^{t-1} | \mathcal{M}^t, \mathcal{P}\right)$ and $p\left(\mathrm{T}_r(\boldsymbol{O}_{\mathrm{C}}^{t-1}) | \mathrm{T}_g(\mathcal{M}^t, \mathcal{P})\right) = p\left(\boldsymbol{O}_{\mathrm{C}}^{t-1} | \mathcal{M}^t, \mathcal{P}\right)$, where $r \in \mathrm{SO}(3), v \in \mathrm{T}(3), r + v = g \in \mathrm{SE}(3)$, then $p(\boldsymbol{x}_{\mathrm{M}}^{t-1} | \mathcal{M}^t, \mathcal{P})$ is SO(3)-equivariant.

*Proof.* According to the convolution formula in probability theory, if $w = u + v$, then

$$p(w) = \int p(u, w - u) du = \int p(w - v, v) dv \quad (16)$$

By using the Eq. (16), we can write every single atom's position density function as

$$
\begin{aligned}
&p\left(\boldsymbol{x}_{\mathrm{M}}^{t-1}|\mathcal{M}^{t},\mathcal{P}\right)\\
=&p\left(\boldsymbol{x}_{\mathrm{C}}^{t-1}+\boldsymbol{x}_{\mathrm{R}}\boldsymbol{O}_{\mathrm{C}}^{t-1}|\mathcal{M}^{t},\mathcal{P}\right)\\
=&\int p\left(\boldsymbol{x}_{\mathrm{C}}^{t-1},\boldsymbol{x}_{\mathrm{M}}^{t-1}-\boldsymbol{x}_{\mathrm{C}}^{t-1}|\mathcal{M}^{t},\mathcal{P}\right)d\boldsymbol{x}_{\mathrm{C}}^{t-1}\\
=&\int p\left(\boldsymbol{x}_{\mathrm{C}}^{t-1}|\mathcal{M}^{t},\mathcal{P}\right)p\left(\boldsymbol{x}_{\mathrm{M}}^{t-1}-\boldsymbol{x}_{\mathrm{C}}^{t-1}|\mathcal{M}^{t},\mathcal{P}\right)d\boldsymbol{x}_{\mathrm{C}}^{t-1}
\end{aligned}
\tag{17}
$$

Since

$$
p\left(\mathrm{T}_g(\boldsymbol{x}_{\mathrm{C}}^{t-1})|\mathrm{T}_g(\mathcal{M}^{t},\mathcal{P})\right)=p\left(\boldsymbol{x}_{\mathrm{C}}^{t-1}|\mathcal{M}^{t},\mathcal{P}\right),
\tag{18}
$$

and

$$
p\left(\mathrm{T}_r(\boldsymbol{O}_{\mathrm{C}}^{t-1})|\mathrm{T}_g(\mathcal{M}^{t},\mathcal{P})\right)=p\left(\boldsymbol{O}_{\mathrm{C}}^{t-1}|\mathcal{M}^{t},\mathcal{P}\right).
\tag{19}
$$

We can obtain that

$$
\begin{aligned}
&p\left(\mathrm{T}_r(\boldsymbol{x}_{\mathrm{M}}^{t-1}-\boldsymbol{x}_{\mathrm{C}}^{t-1})|\mathrm{T}_g(\mathcal{M}^{t},\mathcal{P})\right)\\
=&p\left(\mathrm{T}_r(\boldsymbol{x}_{\mathrm{R}}\boldsymbol{O}_{\mathrm{C}}^{t-1})|\mathrm{T}_g(\mathcal{M}^{t},\mathcal{P})\right)\\
=&\frac{1}{\boldsymbol{x}_{\mathrm{R}}}p\left(\mathrm{T}_r(\boldsymbol{O}_{\mathrm{C}}^{t-1})|\mathrm{T}_g(\mathcal{M}^{t},\mathcal{P})\right)\\
=&\frac{1}{\boldsymbol{x}_{\mathrm{R}}}p\left(\boldsymbol{O}_{\mathrm{C}}^{t-1}|\mathcal{M}^{t},\mathcal{P}\right)\\
=&p\left(\boldsymbol{x}_{\mathrm{R}}\boldsymbol{O}_{\mathrm{C}}^{t-1}|\mathcal{M}^{t},\mathcal{P}\right)\\
=&p\left(\boldsymbol{x}_{\mathrm{M}}^{t-1}-\boldsymbol{x}_{\mathrm{C}}^{t-1}|\mathcal{M}^{t},\mathcal{P}\right)
\end{aligned}
\tag{20}
$$

Therefore, according to Eq. (18), (20), and (17)

$$
\begin{aligned}
&p\left(\mathrm{T}_g(\boldsymbol{x}_{\mathrm{M}}^{t-1})|\mathrm{T}_g(\mathcal{M}^{t},\mathcal{P})\right)\\
=&\int p\left(\mathrm{T}_g\boldsymbol{x}_{\mathrm{C}}^{t-1}|\mathrm{T}_g(\mathcal{M}^{t},\mathcal{P})\right)p\left(\mathrm{T}_{r+v}(\boldsymbol{x}_{\mathrm{M}}^{t-1}-\boldsymbol{x}_{\mathrm{C}}^{t-1})|\mathrm{T}_g(\mathcal{M}^{t},\mathcal{P})\right)d\boldsymbol{x}_{\mathrm{C}}^{t-1}\\
=&\int p\left(\mathrm{T}_g\boldsymbol{x}_{\mathrm{C}}^{t-1}|\mathrm{T}_g(\mathcal{M}^{t},\mathcal{P})\right)p\left(\mathrm{T}_r(\boldsymbol{x}_{\mathrm{M}}^{t-1}-\boldsymbol{x}_{\mathrm{C}}^{t-1})|\mathrm{T}_g(\mathcal{M}^{t},\mathcal{P})\right)d\boldsymbol{x}_{\mathrm{C}}^{t-1}\\
=&\int p\left(\boldsymbol{x}_{\mathrm{C}}^{t-1}|\mathcal{M}^{t},\mathcal{P}\right)p\left(\boldsymbol{x}_{\mathrm{M}}^{t-1}-\boldsymbol{x}_{\mathrm{C}}^{t-1}|\mathcal{M}^{t},\mathcal{P}\right)d\boldsymbol{x}_{\mathrm{C}}^{t-1}\\
=&p\left(\boldsymbol{x}_{\mathrm{M}}^{t-1}|\mathcal{M}^{t},\mathcal{P}\right)
\end{aligned}
\tag{21}
$$

*Proof of Proposition 1.* The sufficiency of SE(3)-invariance of $p(\boldsymbol{s}^{t-1}|\mathcal{M}^{t},\mathcal{P})$ and $p(\boldsymbol{s}^{T})$ is given in **Lemma A.1**, and the sufficiency of SE(3)-equivariance of $p(\boldsymbol{x}^{t-1}|\mathcal{M}^{t},\mathcal{P})$, and SO(3)-equivariance and T(3)-invariance of $p(\boldsymbol{O}^{t-1}|\mathcal{M}^{t},\mathcal{P})$ is given in **Proposition A.2**. Besides, it is easy to obtain that if $p(\boldsymbol{O}^{T})$ and $p(\boldsymbol{x}^{T})$ is SE(3)-invariant distribution, then $p(\boldsymbol{x}_{\mathrm{M}}^{T})$ will be invariant.

## B Model Comparisons.

**With Pocket2Mol and GraphBP.** Pocket2Mol and GraphBP are all auto-regressive models, which violate the physical rules from the perspective of energy[15], while the diffusion models which consider the global interactions are a solution to the problem. Besides, to decide which atom will be added a bond with the next atom, Pocket2Mol needs a classifier to predict the focal atom, so that the training and the prediction are not end-to-end. Finally, D3FG is a functional-group- or fragment-based method, while these two models generate molecules at the atom level.

**With DiffSBDD and TargetDiff.** These two methods are diffusion-based, which considers the global interactions between atoms. In the atom type generation, DiffSBDD uses continuous latent spaces generation, while Targetdiff diffuses the atom types in a uniform distribution. These two models all generate molecules at the atom level, while D3FG generates at the fragment level, and besides the types and positions, the orientations of the functional groups are generated.

**With FLAG.** It is still an auto-regressive model and does not support end-to-end generation since the focal atom needs to be predicted. Besides, we defined our functional groups as rigid bodies with stable intra-structures, while the motifs in FLAG are 2D smiles, with the structures generated by RDKIT. The 3D structures of functional groups are obtained by the training set in D3FG, thus avoiding the problem of distribution shift of FLAG since the training/test motif substructures may not match the RDKIT's generation.

**With DeepFrag.** It is a model for fragment-based lead optimization, in which the protein and the parent is used as condition and the fragment type is predicted by the model. In this way, it is more like an elaboration task defined in D3FG. Here we point out several advantages of D3FG(EHot/Cold) over Deepfrag. First, Deepfrag just predicts the fragment type, without 3D structures, so that the problem of equivariance is not included, but D3FG generates 3D positions of the molecules. Second, although the type is the SE(3)-invariant variable, the model uses CNNs as the model backbone rather than EGNN, so the invariance can also not be preserved. Instead, D3FG assures physical symmetries by using EGNN. Finally, D3FG is a generative model with stochasticity, while Deepfrag only gives the probability of the fragment types, as a discriminative model.

## C  Method Details

**Amino acid context encoding.** Several geometric or type features are embedded to encode amino acids. For the geometric features including torsion angles/dihedrals, and pairwise distances, they are all roto-translational invariant, since the geometric features are all scalars obtained from relative coordinates. Besides, the local coordinates of atoms in a single amino acid are also invariant because it is always fixed in the local frame established by $C_\alpha$, C and N. For the type features including amino acid types, sequential relationships, and pair of amino acid types, the translational and rotational operation is unrelated to them. Thus, the encoded amino acid contexts are roto-translational invariant, leading to the invariance of all the follow-up embeddings.

**Equivariant neural network for linkers.** For the roto-translational equivariance of positions for single atoms, since $\boldsymbol{O}_j^t = \boldsymbol{I}$, Eq. 12 will be written as $G\left(\mathcal{M}^t, \mathcal{P}\right)[j] = \mathrm{MLP}_G(\boldsymbol{h}_j')$, unable to satisfy the equivariance. In this way, we revised it for single atoms by using the EGNN[44] stacked in the final layer for updating the positions, which reads

$$G\left(\mathcal{M}^t, \mathcal{P}\right)[j] = \boldsymbol{x}_j + \frac{1}{C_j} \sum_{i \in \mathcal{I}_{\mathrm{at}} \cup \mathcal{I}_{\mathrm{fg}}} (\boldsymbol{x}_j - \boldsymbol{x}_i)\boldsymbol{h}_i', \tag{22}$$

where we choose $C_j = \|\boldsymbol{x}_j\|_2 + 1$.

## D  Data Preprocessing

**Local frame establishment.** In 3D Euclidian space, for a rigid body including more than mass points that are not co-linear, a local frame can be established. We first choose a center node (center point) $A$ as the origin, and a second node $B$, leading to $\vec{AB}$ as the direction of x-axis. Then, a third node $C$ is selected. By Schmidt orthogonalization of $\vec{AC}$ with respect to $\vec{AB}$, the direction of y-axis can be computed. And finally, the direction of z-axis is obtained by the cross-product of the unit vectors in the x direction and y direction. By this means, the local frame is established by the three nodes, and the other nodes' local coordinates can be determined in the local frame. Because the local frame requires at least three points to establish, the functional groups including only 2 atoms are divided into two linkers.

**Functional group datasets.** We give detailed information on the included functional groups, including 2D graph, 3D structure, time of occurrence in the CrossDocked2020, and the center node (node $A$), node $B$, and node $C$ of the functional group in Table. 10.

Note that in beneze, the symmetric structures lead the frame nodes to be any three consecutive points on the hexagon. Besides, for the functional groups of 'NS(=O)=O' and 'O=CNO', two stable conformations exist, so we in practice regard them as four different types.

Table 5: Frequency of the top ten functional groups that occur most frequently in Crossdocked2020.

| Functional Group | Ref. | Pocket2Mol | TargetDiff | DiffSBDD | D3FG(Joint) | D3FG(Stage) |
|---|---|---|---|---|---|---|
| c1ccccc1 | 0.392 | 0.491 | 0.277 | 0.007 | 0.372 | 0.409 |
| NC=O | 0.147 | 0.075 | 0.142 | 0.201 | 0.082 | 0.107 |
| O=CO | 0.119 | 0.169 | 0.303 | 0.579 | 0.154 | 0.085 |
| c1ccncc1 | 0.045 | 0.072 | 0.049 | 0.018 | 0.027 | 0.052 |
| c1ncc2nc[nH]c2n1 | 0.034 | 0.001 | 0.000 | 0.000 | 0.001 | 0.020 |
| NS(=O)=O | 0.030 | 0.000 | 0.000 | 0.008 | 0.001 | 0.001 |
| O=P(O)(O)O | 0.022 | 0.003 | 0.193 | 0.000 | 0.007 | 0.010 |
| OCO | 0.019 | 0.024 | 0.091 | 0.016 | 0.045 | 0.050 |
| c1cncnc1 | 0.017 | 0.008 | 0.138 | 0.000 | 0.002 | 0.009 |
| c1cn[nH]c1 | 0.016 | 0.011 | 0.001 | 0.000 | 0.003 | 0.004 |
| JSD | - | 0.248 | 0.301 | 0.553 | 0.223 | 0.201 |

Table 6: Ratio of the atoms.

| Atom | Ref. | Pocket2Mol | TargetDiff | DiffSBDD | D3FG(Joint) | D3FG(Stage) |
|---|---|---|---|---|---|---|
| C | 15.866 | 14.956 | 17.744 | 13.526 | 17.766 | 15.999 |
| N | 2.765 | 1.956 | 2.192 | 2.236 | 2.157 | 1.943 |
| O | 4.006 | 2.538 | 4.389 | 3.071 | 3.732 | 3.353 |
| F | 0.309 | 0.084 | 0.239 | 0.160 | 0.193 | 0.170 |
| P | 0.263 | 0.024 | 0.119 | 0.034 | 0.969 | 0.088 |
| S | 0.266 | 0.038 | 0.104 | 0.149 | 0.169 | 0.153 |
| Cl | 0.152 | 0.016 | 0.064 | 0.006 | 0.145 | 0.122 |
| MAE | - | 0.573 | 0.471 | 0.627 | 0.528 | 0.294 |

# E   Experiment Details

**Platform.**   We use a single NVIDIA A100(81920MiB) GPU for a trial. The codes are implemented in Python 3.9 mainly with Pytorch 1.12, and run on Ubuntu Linux.

**Model Details.**   In the diffusion of orientation and position, we employ a cosine variance schedule for $\bar{\alpha}_t$, which reads

$$\bar{\alpha}_t = \cos^2\left(\frac{\pi}{2}(\frac{t}{T} + s)/(1 + s)\right)/\cos^2\left(\frac{\pi}{2}s/(1 + s)\right),\tag{23}$$

where $s = 0.01$. In the diffusion of atom type, $\beta_t$ is set as $\beta_t = \frac{t}{T}$. For the denoiser, the layer number is set as 6, and the embedding size is set as 256. The model is trained with Adam optimizer in 5000 epochs.

**Functional group and Atom type analysis.**   We give a detailed analysis of functional groups and atom types, in Table. 5, 6 and 7.

**Bond Angle and Dihedral analysis.**   Besides bond distance, a detailed analysis of bond angle and dihedral is shown in Table. 8 and  9. It gives the JSD between the reference and the generated molecules, and demonstrates that D3FG generates more realistic drug molecules in comparison with other baselines.

Table 7: Frequency of the atoms.

| Atom | Ref. | Pocket2Mol | TargetDiff | DiffSBDD | D3FG(Joint) | D3FG(Stage) |
|------|------|------------|------------|----------|-------------|-------------|
| C | 0.672 | 0.762 | 0.714 | 0.702 | 0.741 | 0.733 |
| N | 0.117 | 0.100 | 0.088 | 0.116 | 0.124 | 0.089 |
| O | 0.170 | 0.129 | 0.176 | 0.159 | 0.175 | 0.154 |
| F | 0.013 | 0.004 | 0.009 | 0.008 | 0.009 | 0.008 |
| P | 0.011 | 0.001 | 0.005 | 0.002 | 0.002 | 0.004 |
| S | 0.011 | 0.002 | 0.004 | 0.007 | 0.001 | 0.007 |
| Cl | 0.006 | 0.001 | 0.002 | 0.003 | 0.001 | 0.006 |
| JSD | - | 0.098 | 0.059 | 0.054 | 0.075 | 0.056 |

Table 8: JSD on bond angle distributions.

| Angle | Pocket2Mol | TargetDiff | DiffSBDD | D3FG(Stage) | D3FG(Joint) |
|-------|------------|------------|----------|-------------|-------------|
| C-C-C | 0.269 | 0.272 | 0.304 | **0.255** | 0.258 |
| C-C-N | 0.254 | 0.267 | 0.313 | 0.256 | **0.255** |
| C-N-C | 0.286 | **0.241** | 0.319 | 0.269 | 0.277 |
| C-C-O | 0.317 | 0.295 | 0.345 | **0.293** | 0.295 |
| C-O-C | 0.308 | 0.311 | 0.372 | **0.310** | 0.314 |
| C-N-N | 0.294 | 0.276 | 0.301 | **0.270** | 0.281 |
| N-C-O | 0.300 | 0.295 | 0.326 | **0.282** | 0.291 |
| N-C-N | 0.304 | 0.288 | 0.342 | **0.282** | 0.292 |

Table 9: JSD on dihedral distributions.

| Dihedral | Pocket2Mol | TargetDiff | DiffSBDD | D3FG(Stage) | D3FG(Joint) |
|----------|------------|------------|----------|-------------|-------------|
| C-C-C-C | 0.151 | 0.149 | 0.158 | 0.141 | **0.138** |
| C-C-C-N | 0.176 | **0.165** | 0.224 | 0.169 | 0.175 |
| C-C-C-O | 0.183 | 0.159 | 0.206 | **0.156** | 0.164 |
| C-C-O-C | 0.180 | 0.174 | 0.231 | 0.167 | **0.149** |
| C-C-N-C | 0.165 | 0.142 | 0.223 | **0.136** | 0.146 |
| C-C-N-O | 0.277 | 0.270 | 0.285 | **0.264** | 0.293 |
| C-N-C-O | 0.453 | 0.430 | 0.398 | 0.358 | **0.335** |
| N-C-C-O | 0.315 | 0.253 | 0.303 | **0.244** | 0.272 |
| C-N-C-N | 0.340 | 0.317 | 0.328 | **0.254** | 0.263 |

Table 10: The included functional groups in D3FG. 'T' is the occurrence times of the functional group in the datasets (100,000 ligands).

| Smiles | 2D graph | 3D structures | A | B | C | T |
|--------|----------|---------------|---|---|---|---|
| c1ccccc1 | | | 1 | 0 | 2 | 131148 |
| NC=O | | | 1 | 0 | 2 | 49023 |
| O=CO | | | 1 | 0 | 2 | 39863 |
| c1ccncc1 | | | 3 | 2 | 4 | 15115 |
| c1ncc2nc[nH]c2n1 | | | 7 | 3 | 6 | 11369 |
| NS(=O)=O | | | 1 | 0 | 2 | 10121 |
| O=P(O)(O)O | | | 1 | 0 | 2 | 7451 |
| OCO | | | 1 | 0 | 2 | 6405 |
| c1cncnc1 | | | 3 | 2 | 4 | 5965 |
| c1cn[nH]c1 | | | 2 | 3 | 1 | 5404 |

| Smiles | 2D graph | 3D structures | A | B | C | T |
|---|---|---|---|---|---|---|
| O=P(O)O | | | 0 | 1 | center(2,3) | 5271 |
| c1ccc2ccccc2c1 | | | 3 | 2 | 4 | 4742 |
| c1ccsc1 | | | 3 | 2 | 4 | 4334 |
| N=CN | | | 1 | 0 | 2 | 4315 |
| NC(N)=O | | | 2 | 1 | 3 | 4167 |
| O=c1cc[nH]c(=O)[nH]1 | | | 7 | 1 | 5 | 4145 |
| c1ccc2ncccc2c1 | | | 3 | 2 | 4 | 3519 |
| c1cscn1 | | | 2 | 3 | 1 | 3466 |
| c1ccc2[nH]cnc2c1 | | | 5 | 4 | 6 | 3462 |
| c1c[nH]cn1 | | | 3 | 2 | 4 | 2964 |
| O=[N+][O-] | | | 1 | 0 | 2 | 2702 |

| Smiles | 2D graph | 3D structures | A | B | C | T |
|--------|----------|---------------|---|---|---|---|
| O=CNO |  |  | 1 | 0 | 2 | 2477 |
| NC(=O)O |  |  | 1 | 0 | 2 | 2438 |
| O=S=O |  |  | 1 | 0 | 2 | 2375 |
| c1ccc2[nH]ccc2c1 |  |  | 3 | 4 | 2 | 2301 |

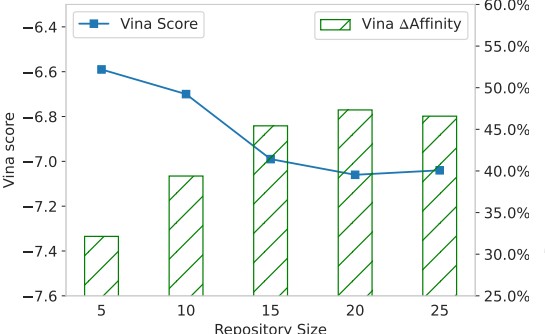

| Size | 5 | 10 | 15 | 20 | 25 |
|---|---|---|---|---|---|
| QED | 0.489 | 0.484 | 0.502 | 0.496 | 0.501 |
| SA | 0.821 | 0.814 | 0.836 | 0.843 | 0.840 |
| LogP | 2.774 | 2.795 | 2.802 | 2.759 | 2.821 |
| Lipinski | 4.910 | 4.983 | 4.937 | 4.931 | 4.965 |

Table 11: Metrics of chemical properties with the change of repository sizes

Figure 5: Affinity metrics with the change of repository sizes.

