# OpenReview forum: "Functional-Group-Based Diffusion for Pocket-Specific Molecule Generation and Elaboration"
_NeurIPS.cc/2023/Conference — NeurIPS 2023 poster_

### Official Review · Reviewer_6gZx · 2023-06-17

**Soundness:** 3 good
**Presentation:** 3 good
**Contribution:** 1 poor
**Rating:** 4
**Confidence:** 5

**Summary:**

In this paper, a functional-group-based diffusion model called D3FG is proposed to generate molecules in 3D for target protein binding. Two generation schemes including joint and two-stage generation schemes are formulated.

**Strengths:**

1. The paper is well-written and easy to follow.
2. It is the first functional-group-based diffusion model for structure-based drug design.

**Weaknesses:**

1.  The performance improvement over previous methods is limited. For example, in table.2, TargetDiff achieves the lowest JS divergence on most bond distances. In Table. 4, TargetDiff achieves the best performance on the vina score. In Figure. 3 the D3FG does not show an advantage over other baselines with respect to atom type distribution.
2. The evaluations are not comprehensive. For example, besides divergence between bond distances, bond angles and dihedral angles should be evaluated. The influence of the size of the functional group set is not explored.
3. The technical contribution is limited. The application of diffusion models for drug design is not new. It seems the required techniques are proposed in previous works including TargetDiff [7], DiffSBDD [8], and DiffAB [52].
4. Some related works are not discussed. For example, some previous methods also leverage functional groups or motifs for molecule design: JT-VAE [1], PS-VAE [2], and FLAG [3].

[1] Jin et al., Junction tree variational autoencoder for molecular graph generation, ICML 2018
[2] Kong et al., Molecule generation by principal subgraph mining and assembling, NeurIPS 2022
[3] Zhang et al., Molecule generation for target protein binding with structural motifs, ICLR 2023

**Questions:**

1. It seems that the reported scores in Table. 4 are different from the original papers of baselines. Could the authors explain that?

**Limitations:**

Yes, the authors have adequately discussed the limitations.

---

> ### Author Rebuttal · Authors · 2023-08-06
>
>
> Thanks for your advice, and we add more experimental details in the Appendix E according to your advice as shown in CQ2 in General Response. Here is the response to your doubts.
>
> Response to W1: On one hand, the improvements in docking scores are not significant, according to Table.4, but the other chemical properties of generated molecules are overall the best. On the other hand, D3FG can generate more realistic drug-like molecules, with complex substructures (Table. 1 in paper). On atom type distribution, it achieves the lowest MAE. Besides, on the other geometries like bond length, bond angle, and dihedral distribution, it shows the overall best performance, according to the Table.2 in the paper and Table.2 and 3 in the CQ2 in General response.
>
> Response to W2: We hope CQ2 in General Response can remove your doubts and change your opinion.
>
> Response to W3: We have discussed the technical contributions of D3FG over TargetDiff and DiffSBDD in the paper, and concluded them as three points. (i) Functional-group-based method, so that more complex substructures with pharmacodynamic function can be generated. (ii) SO(3) Diffusion, to generate the orientation of functional groups regarded as rigid bodies. (iii) Fragment-linker designation, with two solutions whose  effectiveness is validated through experiments.  The of two-stage schemes, which is similar to classical CADD, perform better. Our contributions over DiffAB can be concluded as: (i) Establish a repository consisting of a functional group with stable substructures, rather than amino acids which have been completely defined so that positions, orientations, and types can be generated through diffusion models. (ii) Consider single atoms as linkers, symmetries of which IPA cannot handle, and use EGNN as another head to generate positions. (iii) Employ Bert-style diffusion (D3PM) rather than uniform diffusion on categorical variables.
>
> Response to W4: We add these methods into related works as shown in CQ3, and hope it helps to enhance the integrity of the article and changes your opinion. the code is open to the public with a bug fixed on May 10, 2023. When we cannot have insights through the source code, details are hard to be provided.  Besides,here we give comparisons with FLAG. (i)It is an auto-regressive model, violating the physical rules from the perspective of energy[5], while the diffusion models which consider the global interactions are solutions. (ii)It needs a classifier to decide which atom will be added a bond with the next motif, so the training and the prediction are not end-to-end. In contrast, D3FG generates molecules with a diffusion model as long as the number of nodes is given. (iii) Finally, functional groups are rigid bodies in D3FG, while the motifs in FLAG are 2D smiles (https://github.com/zaixizhang/FLAG/blob/main/utils/vocab.txt), with the structures generated by RDKIT(Line. 188, 189 in motif_sample.py in the same link). The 3D structures of functional groups are obtained by the training set in D3FG, thus avoiding the problem of distribution shift of FLAG since the training/test motif substructures may not match the RDKIT’s generation, which has been discussed in [6]. Besides, since the pre-trained model of FLAG is not provided, we cannot give a detailed experimental comparison.
>
> Response to Q1:  All the metrics are reported based on our reproduction of the methods, but there are unavoidable differences from the original papers. We think there are two main reasons:
> - (i) Different platform. We take Pocket2Mol as examples. As you can see, the original test platform is 'NVIDIA V100 GPUs with Python 3.8 and Pytorch 1.9.0 a', while in our experiments, it is ' NVIDIA A100(81920MiB) GPU with Python 3.9 and Pytorch 1.12.'. The CUDA version, pytorch version and many environmental variables affect the test results. However, once the protocol is unified on the same one, we can ensure the rigor of experiments.
> - (ii) Stochasiticity. All these methods are generative models, so that randomness is unavoidable.
>
> Overall, the differences are acceptable. The vina docking score of Pocket2Mol is reported as -7.288 originally and -7.05 in DiffSBDD, and in our experiments, it is -6.92, of minor and acceptable deviation.

---

> > ### Comment · Reviewer_6gZx · 2023-08-13
> > **Thanks for the response**
> >
> > Thanks for the authors' response! However, most concerns still exist (e.g., the limited technical contribution and obviously lower baseline performance compared to original papers). The source code is also not provided. Therefore, the reviewer leans towards rejection.

---

> > > ### Author Response · Authors · 2023-08-14
> > > **Thanks for this timely reply.**
> > >
> > > In the rebuttal, we describe our contributions over the previous works in W3, and the improvements of D3FG on realistic generation over previous methods in W1 and W2. If there is any doubts on these two points, please feel free to point out them and let us to explain them.
> > >
> > > Besides, for further reproduction, we update our code in a an anonymous repository ( https://anonymous.4open.science/r/D3FG-D1FC/ ), and hope it can help to change your opinion towards a more positive way.
> > >
> > > Thanks a lot for your reply.

---

> > > > ### Comment · Reviewer_6gZx · 2023-08-16
> > > > **Thanks for your reply!**
> > > >
> > > > Thanks for providing the code and feedback! However, most of the concerns remain (see below). Therefore, the reviewer decides to keep the score.
> > > >
> > > > W1: In Table. 4, TargetDiff achieves the best performance on the vina score. In Figure. 3 the D3FG does not show an advantage over other baselines with respect to atom type distribution. Moreover, some reported baseline metrics are obviously lower than the original papers.
> > > >
> > > > W2: The author only explores the influence of the size of the functional group set from 5 to 25. What about more functional groups, say 100?
> > > >
> > > > W3: Most of the architectural components of D3FG have been shown in previous papers. D3FG puts them together in an effective way. But the technical ideas are not novel.
> > > >
> > > > W4: The related works are not compared, such as FLAG.

---

> > > > > ### Author Response · Authors · 2023-08-17
> > > > > **Response**
> > > > >
> > > > > Thank you for your prompt response. We have the following explanation for your concern:
> > > > >
> > > > > W1: ( Does D3FG achieve state-of-the-art performance? ) While TargetDiff narrowly edged out D3FG in the Vina Score, D3FG demonstrated the best performance in the other docking score called Gnina Score (in the third and fourth column in Table 4). Given that these scoring functions are approximation to real-world energy  by the algorithms, we cannot say determinately that TargetDiff demonstrates better performance. However, it can be concluded that D3FG is the current SOTA method that possesses performance comparable to the existing methods that demonstrate the best performance like TargetDiff and Pocket2Mol. Besides, in the analysis in realistic generation, the comparison of the number of atoms is only one aspect, demonstrating the competitive performance of D3FG, and in functional group generation comparison (Table 1) and  in geometry comparisons like the bond length (Table 2), bend angle (Table 2 in General Response) and torsional angle (Table 3 in General Response), D3FG also shows the overall best performance. We suppose that it is hard to ask for one method to perform best in every aspect, but in the overall evaluation, D3FG is one of the most effective SBDD methods.
> > > > >
> > > > > W2: ( Why 25 functional groups? ) As you can see in Table 8 in the Appendix, the `c1ccc2[nH]ccc2c1’ functional group is the 25th ones, only 2.3% of molecules have the substructures, so we assure the 25 functional groups are enough to consist of a repository and cover most generation tasks in Crossdocked. Besides, we define the functional groups as the fragments with pharmacological functions and stable structures, so we have to check the stability of the defined functional groups in the datasets one by one, and extend to more types if there are more than one stable conformation in the functional groups. The repository is extendable, but extension to a larger one takes a lot of time. There, in the Conclusion and Feature Work, we recognize that such a repository is only a prototype and will continue to expand in the future to make it more generalized.
> > > > >
> > > > > W3: ( Novelty of D3FG ) We have explained it in the *Response to W3* in our rebuttal. Although the novelty of the methodology are usually controversial, we hope the reviewers can focus more on our contributions over the previous methods.
> > > > >
> > > > > W4: ( More comprehensive comparison ) Different from TargetDiff, DiffSBDD and Pocket2Mol, the pre-trained model of FLAG is not provided, so it is very hard for us to give a detailed experimental comparison. If there are other effective open source methods that are not included in the comparison, please feel free to suggest them and we will be happy to include them in the comparison experimentally.

---

### Official Review · Reviewer_Y6qT · 2023-06-30

**Soundness:** 4 excellent
**Presentation:** 4 excellent
**Contribution:** 3 good
**Rating:** 7
**Confidence:** 4

**Summary:**

Pocket-specific molecule generation has received considerable attention in recent years, and the authors propose a functional-group-based diffusion model to address this task. The model considers the generation of complete molecules as the assembly of functional groups (fragments) and atoms that connect these functional groups. They employ a diffusion-based generative scheme to determine the coordinates, orientation, and types of these fragments. The experiments demonstrate the satisfactory generation performance of the model and its potential applications in molecule elaboration.

**Strengths:**

1. The paper is well-written, and the narrative is clear and concise.
2. The authors address many important details, including the determination of functional groups, handling chirality, and generating heterogeneous graphs.
3. Overall, while the idea of decomposing molecules is not novel, the authors provide a robust model and conduct experiments to validate its superiority.

**Weaknesses:**

1. In the Method section, it would be beneficial to provide an overview of the entire method before delving into the specifics of each part.
2. The aromatic elements C:C and C:N in Table 2 could be written in lowercase to maintain consistency with convention.
3. It would be beneficial to include more visualizations (2D/3D) of generated molecules.

**Questions:**

1. In the Introduction, it is mentioned that protein amino acids are represented as functional groups and linking atoms (Line 42). However, in the Method section, the amino acid space I_aa is preserved (Line 60). Why is the amino acid space still necessary if any node can be considered either a functional group or atom?
2. The functional groups are treated as rigid fragments, and their orientations are predicted. But what if the functional groups have internal rotatable bonds and cannot be regarded as rigid (e.g., O=CNO)?  Additionally, how is the symmetry of functional groups considered when defining orientation? For example, a benzene ring can have multiple orientations resulting in the same arrangement of the six atoms. Which orientation is chosen during training?
3. In the experiments, why did you use EFGs to segment molecules instead of BRICS composition, as done in many previous fragment-based works? Why did you choose 25 functional groups? How does this choice cover the total number of atoms in a molecule, i.e., the ratio of atoms in functional groups to the total number of atoms for a molecule? Did you consider other sizes for the set of functional groups?
4. Does the node number include the protein in Line 220?
5. For the molecule elaboration experiment, did you retrain the model specifically for this task, or did you use the same model as in the molecule generation experiment?
6. In Line 288, what does the word "them" refer to? It seems ambiguous.
7. Why does D3FG(EHot) generate molecules with higher affinities? In my opinion, since functional groups with high binding contributions were removed, it is likely that the generated functional groups have weaker binding, resulting in new molecules with weaker affinities. On the other hand, in the D3FG(ECold) experiments, a functional group with higher binding contribution is more likely to be generated, replacing the weaker functional group and leading to higher affinities. Why do the experimental results not align with this explanation?

**Limitations:**

The authors have addressed the limitations.

---

> ### Author Rebuttal · Authors · 2023-08-06
>
> We sincerely thank you a lot for your appreciation of the work, and the advice with deep insights. Here is the response to your concerns.
>
>
> Response to Q1. The amino acid index set is preserved because, in the diffusion process, only the linkers and functional groups are added with noise, so the index sets of the three types of nodes are defined for differentiating them. Besides, in description of the amino acid embedding, the index set is also used.
>
> Response to Q2. By using EFGs to decompose the molecules, we find that the functional groups with high frequency usually have stable structures. In detail, we randomly choose one functional group's structures which are split from one molecule, and match all the other functional groups with the same smiles but from other molecules in 3D. We define the substructures are stable if the RMSD is less equal than 0.005A. When there are RMSE larger than the threshold, we pick all of them and repeat the process. In this way, we find the functional groups are stable in 3D, since most of them only have one stable structure, and only two have multiple structures. The orientation is defined as a global vector that is the z-axis in the local frame. For example, in a benzene ring, the six atoms are coplanar, so no matter how the local frame is built, the z-axis is the normal vector to that face, and this normal vector serves as a representation of the rotational orientation.
>
> Response to Q3. Our D3FG method is adaptable and can be applied to various fragment databases. Although technically the selection of EFGs and BRICS can be interchangeable, the EFGs method aligns better with our objectives. Unlike BRICS, which deconstructs molecules into synthesizable building blocks, the EFGs method provides a viable approach to obtaining a series of chemical fragments. Reports have shown that the number of fragments derived from BRICS increases linearly with the size of the molecular library[1]. Conversely, the fragments from EFGs are manually curated, thus providing a robust representation of molecular diversity. These fragments effectively cover drug-like chemical space and have demonstrated efficacy in distinguishing between inhibitors and non-inhibitors[2]. Therefore, we opted to use EFGs in this work, and shifting to the BRICS setting is straightforward and can be done according to the user's preference. Besides, in choosing the functional groups, we aim to establish a repository of functional groups that most molecules can be decomposed in. As you can see in Table 8 in the appendix, the `c1ccc2[nH]ccc2c1’ functional groups are the 25th ones, only 2.3% of molecules have the substructures, so we assure the 25 functional groups are enough to consist of a repository and cover most generation tasks in Crossdocked. In detail, we test the sensitivity of performance affected by repository size in CQ2 in General response. We hope it can remove your doubts.
>
> Response to Q4. Yes, it includes, since D3FG and DiffSBDD represent protein pockets at amino acid levels, the node numbers are much lower than TargetDiff.
>
> Response to Q5. The model is retrained, since the contextual information changes. In molecule generation, the condition is the protein structure, and in the elaboration task, it is the protein and the remaining molecule's fragments.
>
> Response to Q6.  'Them' refers to the 'removed functional groups with the highest scores'. we are sorry that the confusion made by our presentation, and will change the description in the next version.
>
> Response to Q7. As the pharmacophoric sites are usually fixed in a pocket, and the functional groups with high hotmap scores usually lie in the sites. By replacing it with other fragments with different local structures and orientations, the likelihood of enhanced interactions with proteins is higher. However, when removing the functional groups with low scores, the generated functional groups usually lies in the 'cold' sites as a result of energy constraints within the molecule, so optimizing this part of fragments often contributes very little to the binding affinity.
>
> [1] HierS: hierarchical scaffold clustering using topological chemical graphs. Journal of medicinal chemistry
>
> [2] Fingerprint-based in silico models for the prediction of P-glycoprotein substrates and inhibitors

---

> > ### Comment · Reviewer_Y6qT · 2023-08-14
> > **Thanks for your reply.**
> >
> > I appreciate your informative reply. Your thorough explanation has indeed clarified my inquiries. However, it appears that the weakness of the paper mentioned in the review has not been examined or discussed yet.

---

> > > ### Author Response · Authors · 2023-08-15
> > > **Response.**
> > >
> > > Thanks for your reply, and we sincerely appreciate your recognition of our work.
> > > We have adopted your advice and will update the D3FG in the next version.
> > > However, since the revision of paper is not allowed to be uploaded, we list the improvements of our paper of the revised version according to your advices, which is not uploaded.
> > >
> > > For W1. We added an overall description of D3FG in Section 4, as
> > >
> > > D3FG firstly decomposes molecules into two categories of components: functional groups and linkers, and them use the diffsion generative model to learn the type and geometry distributions of the components.
> > > In this section, we describe the D3FG by four parts: (i) the diffusion model as the generative framework, in which the three variables are generated; (ii) the denoisers parameterized by graph neural networks, satisfying certain symmetries so that the generative model is SE-(3) invariant; (iii) the sampling process in which the molecules are generated by the trained models; (iv) the further problems resulted from heterogenous graph with two solutions.
> > >
> > > For W2. The notation is changed according to your suggestions.
> > >
> > > For W3. More generated protein-molecules pairs are added in the Appendix E, with 2D and 3D graphs of molecules are attached.

---

### Official Review · Reviewer_sFPS · 2023-07-06

**Soundness:** 2 fair
**Presentation:** 2 fair
**Contribution:** 2 fair
**Rating:** 3
**Confidence:** 4

**Summary:**

This paper proposes a so-called functional-group-based diffusion generative model, namely D3FG, to generate molecules with realistic substructures conditioned on the protein binding sites. D3FG represents the protein-ligand docking system as a fragment-based system. The molecule fragments are molecule substructures and the protein fragments are amino acids. Thus, the original protein-ligand docking system is reduced to a coarsen graph with molecule substructures and amino acids as nodes. The molecule substructures are connected with heavy atoms as linkers. Thus, the denoising and diffusion processes are defined with molecule substructure translation probabilities. Although this idea is interesting, the reviewer is concerned about the paper presentation, literature reviewer, and marginal performance improvement.

**Strengths:**

Representing the protein-molecule docking system as the coarsened graph reduces the computation expenses of the diffusion generative models. Recently, this scheme has drawn much attention and there are some related work has already shown the efficiency of this scheme. Moreover, molecule elaboration is also related to ligand optimization, which aim to optimize the ligand efficiency by modifying it structure.

**Weaknesses:**

1. Limited Novelty and Insufficient literature review. This paper represents the fragment-based method for ligand generation and claims no prior work employs the fragment-based generation scheme. However, there are already some methods that employ fragment-based generation. For example, Deepfrag [1], FLAG [2], and SQUID [3] are all fragment-based generative methods for target-aware molecule design. It is necessary to discuss the difference and highlight the contributions.

2. The paper presentation needs to be improved. For example, what is the intuition of the definition of absorbing state in Line 105? What is the connection between the BERT-style objective and the denoising objective in Eq.4? What are the different levels of nodes in Section 4?

3. Definition of "functional group". The functional group often refers to the molecule substructures that contribute most to the chemical properties. However, as shown in Section 5.1, the functional group in this paper refers to the most frequent substructures. Mining the most frequent substructures for molecule generation is not new and has been employed in [2,4].

4. Marginal empirical improvement. Moreover, there is only marginal empirical improvement over the baseline methods as shown in Table 4.

5. Unfair comparison. Since the proposed method already employs the most frequent substructures for model training, it is unfair to choose the "'Ratio’ of the top ten functional groups with the highest frequency" as an evaluation metric in Table 1.


[1]. Deepfrag: a deep convolutional neural network for fragment-based lead optimization. Chemical Science.

[2]. Molecule generation for target protein binding with structural motif. ICLR 2023.

[3]. Equivariant shape-conditioned generation of 3d molecules for ligand-based drug design. ICLR 2023.

[4]. Molecule Generation by Principal Subgraph Mining and Assembling. NeurIPS 2022.



**Questions:**

The Questions are listed in the Weaknesses Section above.

---

> ### Author Rebuttal · Authors · 2023-08-06
>
>
> We thank you a lot for your constructive advice and answer your questions one by one, as below.
>
> Response to Q1:
> We add the three methods to the related work as discussed in CQ3 in General Response. In detail, the novelty of D3FG and its differences are listed below:
>
> - We did not compare FLAG[2] in method and experiment because the code is open to the public with a bug fixed on May 10, 2023. When we cannot have insights through the source code, details are hard to be provided. Here we give comparisons with FLAG. (i)It is an auto-regressive model, violating the physical rules from the perspective of energy[5], while the diffusion models which consider the global interactions are solutions. (ii)It needs a classifier to decide which atom will be added a bond with the next motif, so the training and the prediction are not end-to-end. In contrast, D3FG generates molecules with a diffusion model as long as the number of nodes is given. (iii) Finally, functional groups are rigid bodies in D3FG, while the motifs in FLAG are 2D smiles (https://github.com/zaixizhang/FLAG/blob/main/utils/vocab.txt), with the structures generated by RDKIT(Line. 188, 189 in motif_sample.py in the same link). The 3D structures of functional groups are obtained by the training set in D3FG, thus avoiding the problem of distribution shift of FLAG since the training/test motif substructures may not match the RDKIT’s generation, which has been discussed in [6]. Besides, since the pre-trained model of FLAG is not provided, we cannot give a detailed experimental comparison.
>
> - For Deepfrag[1], it is a model for fragment-based lead optimization, in which the protein and the parent is used as condition and the fragment type is predicted by the model. So it is more like an elaboration task defined in D3FG. Here are several advantages of D3FG(EHot/Cold) over Deepfrag. (i)Deepfrag just predicts the fragment type, without 3D structures, so that the problems of symmetry are not included, but D3FG generates 3D positions of the molecules. (ii)Although the type is SE(3)-invariant, the model uses CNNs as the backbone rather than EGNN, so the invariance cannot be preserved. Instead, D3FG assures physical symmetries by using EGNN. (iii)D3FG is a generative model with stochasticity, while Deepfrag only gives the probability of the fragment types, as a discriminative model.
>
> - For SQUID[3], it is a shape-conditioned molecule generation, in which the shape is defined as a point cloud as the outline of the molecule’s 3D structure, while our task is pocket-specific molecule generation, where the shape of the pocket may not be similar as the outline of molecules. In the experiments, we cannot find a similar task to SBDD in SQUID. Therefore, the only common idea of D3FG and SQUID lies in `Fragment-based’. To tell the novelty of ours, SQUID is still an auto-regressive model, similar to Pocket2Mol, with the focal atom prediction modules, different from our end-to-end diffusion model. And we use SO(3) diffusion to generate the orientation of fragments.
>
> Response to Q2:
> The absorbing state and BERT-style training objective is given in the reference[8] (D3PM) in the new version, we are sorry that we miss the reference. The nodes diffused into an absorbing state can be regarded as masked linker/functional groups, so optimizing a cross-entropy defined in Eq.(4) is like pre-training BERT.  As discussed in Sec.4.4, functional groups are regarded as rigid bodies, which occupy a certain volume of space, while linkers are mass points, so the nodes are of different levels: Nodes of functional groups are higher-level nodes consisting of atoms as lower-level nodes.
>
> Response to Q3:
> We use EFGS[7] to decompose the molecules, in which the substructures are defined as ‘a group of atoms that has similar chemical properties whenever it occurs in different compounds’, rather than the general defined motifs with substructures that vary with the datasets. We did not aim to ’MINE’ the frequent substructures. Instead, we try to establish a repository of functional groups that most molecules can be decomposed in, and use the repository to generate or elaborate the new molecules. We suppose that the manual establishment of such a repository with both 2D and stable 3D structures is one of our contributions.
>
> Response to Q4:
> We admit that improvements are not significant in the binding scores compared with TargetDiff, which is one of D3FG’s limitation, but other chemical properties show advantages. Besides, we argue that D3FG generates more realistic molecules with complex substructures from Table.1 and Table.2 and 3 in the General Response.
>
> Response to Q5:
> First, it is our motivation of generating complex functional groups as substructures in molecules (Line 32, 33), so it is natural to compare the generated complex substructures of different methods to show that D3FG fixes the problem. Secondly, even if the complex substructures are generated, they may not be assembled as a valid molecules, because they occupy a large space, and lead to linker-fragment intersection. Therefore, the problem is not trivial, and D3FG(Stage) is a good solution, by regarding the graph as heterogenous and using two-stage generation scheme.
>
> [1] Deepfrag: a deep convolutional neural network for fragment-based lead optimization. Chemical Science.
>
> [2] Molecule generation for target protein binding with structural motif. ICLR 2023.
>
> [3] Equivariant shape-conditioned generation of 3d molecules for ligand-based drug design. ICLR 2023.
>
> [4] Molecule Generation by Principal Subgraph Mining and Assembling. NeurIPS 2022.
>
> [5] Diffbp: Generative diffusion of 3d molecules for target protein binding, 2022
>
> [6] Torsional Diffusion for Molecular Conformer Generation, 2022
>
> [7] Extended Functional Groups (EFG): An Efficient Set for Chemical Characterization and Structure-Activity Relationship Studies of Chemical Compounds. 2015
>
> [8] Structured denoising diffusion models in discrete state-spaces, 2023

---

> > ### Author Response · Authors · 2023-08-20
> > **Reply.**
> >
> > Thank you for your valuable time and constructive suggestions.
> >
> > Considering that the author-reviewer discussion phase is nearing the end, we would like to be able to confirm whether our responses have addressed your concerns.
> >
> > If there is anything else that is not clear, please feel free to contact us.
> >
> > Best,
> >
> > Authors

---

### Official Review · Reviewer_i3fG · 2023-07-07

**Soundness:** 3 good
**Presentation:** 3 good
**Contribution:** 3 good
**Rating:** 6
**Confidence:** 4

**Summary:**

This paper proposes to generate 3D molecules using functional groups and linker atoms in a diffusion model. Using functional groups as building blocks help the model to generate realistic local structures. In the proposed diffusion model, the atom/functional group type, coordinates and orientation are predicted.  In addition, the authors experimented with two strategies, joint and two-stage, to generate heterogenous graph, which contains functional groups and linker atoms. The performance indicates the proposed method achieves SOTA performance in structure-based drug design task.

This paper also proposed a new molecular elaboration task and curated a dataset for this task.

**Strengths:**

1. This paper implement functional groups in diffusion-based structure-based drug design model for the first time.
2. The performance indicates using functional groups as building blocks can improve QED and SA, which have been a challenge for previous diffusion models.
3. This paper proposes a new task, molecule elaboration, for structure-based drug design and provides a dataset for it.

**Weaknesses:**

The paper proposed a new molecule elaboration task, and some existing baselines (e.g. Pocket2Mol[1], 3DSBDD[2], TargetDiff[3] and DiffSBDD[4]) and can be easily adapted for this task. Currently, there is no performance comparison for molecule elaboration task, and it is recommended for the authors to evaluate their model with existing baselines which can be used for this task.


[1] Peng, Xingang, et al. "Pocket2mol: Efficient molecular sampling based on 3d protein pockets." ICML 2022.

[2]Luo, Shitong, et al. "A 3D generative model for structure-based drug design." NeurIPS 2021.

[3] Guan, Jiaqi, et al. "3d equivariant diffusion for target-aware molecule generation and affinity prediction." ICLR 2023.

[4]Schneuing, Arne, et al. "Structure-based drug design with equivariant diffusion models." arXiv preprint arXiv:2210.13695 (2022).

**Questions:**

1. How many percentage of the generated molecules are complete (all the generated atoms are connected into one molecule)?
2. In molecular elaboration task, is the remaining fragments included in D3FG as a condition for elaboration? If not, can the proposed model guarantee that the newly generate fragment is suitable to be connected with the remaining fragments?
3.  Why does the two-stage approach consistently outperform the joint training one? Is there any explanation?

---

> ### Author Rebuttal · Authors · 2023-08-06
>
>
> We sincerely thank you a lot for your appreciation of the work, and the advice with deep insights. Here is the response to your concerns.
>
> Response to comparison to baselines for molecule elaboration:
> As the model for molecule elaboration task in 3D is scarce, we here use STRIFE [1] (auto-regressively-vae) as a reasonable alternative since the STRIFE also extracts the pharmacophoric information with hotspots map. However, the model only elaborates molecules in 2D, so we firstly use the re-trained models to generate 2D molecules conditioned on protein’s structural information and the large fragments without the hottest/coldest functional groups, then use MMFF in RDKit to obtain the conformation, so that the docking score can be calculated.  The metrics on metrics are presented below:
>
> |             | Vina Score | Vina Delta Aff | Gnina Score | Gnina Delta Aff | QED   | SA    | LogP  | Lipinski |
> |-------------|------------|----------------|-------------|-----------------|-------|-------|-------|----------|
> | STRIFE(Hot)  | -7.01      | 42.86%         | 5.13        | 33.35%          | 0.483 | 0.722 | 0.811 | 4.220    |
> | STRIFE(Cold) | -6.96      | 40.20%         | 5.14        | 34.49%          | 0.479 | 0.724 | 0.809 | 4.225    |
> | D3FG(EHot)  | -7.19      | 51.78%         | 5.51        | 56.53%          | 0.482 | 0.731 | 0.814 | 4.330    |
> | D3FG(ECold) | -7.02      | 44.03%         | 5.16        | 32.69%          | 0.476 | 0.707 | 0.820 | 4.228    |
>
>
>
> It shows that in the four chemical properties, the differences are very small. However, in the docking score evaluation, the differences are significant. One of the suspectable reasons is that STRIFE elaborates the molecule auto-regressively at the atom level, and thus the generated fragments in the binding sites are usually small and cannot form complete functional groups with pharmacodynamic function because the problem of early-stopping and unrealistic substructures in the auto-regressive method [2]. And thus, since the D3FG generates larger fragments in the binding sites, interactions are more likely to occur [2, 3].
>
>
> Response to Q1.
> Statistically, in D3FG(stage) 87.16% molecules are generated with all bonds connected, so the restricted validity is 87.16%. In practice, when there are disconnected bonds, we will select the largest fragments as the molecules as DiffSBDD does, and preserve it if the size is larger than 80% of the generated ones. In this way, the non-strict validity is 96.64%. However, in D3FG(joint), the two metrics are only 71.42% and 92.37%.  We add the metrics in the Appendix.
>
> Response to Q2.
> In the elaboration task, both the pocket and the remaining fragments are all used as contextual information. The model is re-trained on this task, and it only generates one functional group with its type, position and orientation.
>
> Response to Q3.
> We think that in the binding system, the functional groups are considered as the fragment level, and the linkers are at the atom level. In section 4.4, we give a brief description of these two levels of nodes. To be specific, in the two-stage scheme, the first diffusion model will locate the pharmacophoric sites and fill the appropriate functional groups into it, and then, the second diffusion model with the other GNN as its denoiser will use the linkers to link them, which is conditioned on pocket and functional groups' structure. Since functional groups are rigid bodies that occupy a certain space, the second model learns that the linkers should not intersect into the generated fragments, so the validity is better, and the generated molecules are more realistic than the ones generated by joint scheme, in which the linkers and functional groups are all regarded at the same level, and denoised by a single GNN.
>
> [1] Incorporating Target-Specific Pharmacophoric Information into Deep Generative Models for Fragment Elaboration. J Chem Inf Model. 2022 May
>
> [2] Diffbp: Generative diffusion of 3d molecules for target protein binding, 2022
>
> [3] 3d equivariant diffusion for target-aware molecule generation and affinity prediction. In International Conference on Learning Representations, 2023

---

> > ### Comment · Reviewer_i3fG · 2023-08-18
> > **Thanks for the response**
> >
> > Thanks for the reply and all my questions have been clarified.
> > The authors provide both strict and non-strict validities for D3FG. Please also include the validities of other baselines when updating the Appendix. It is more clear after the authors explain the settings and details of molecular elaboration task, it would be helpful to include these in the final manuscript as well.

---

> > > ### Author Response · Authors · 2023-08-20
> > > **Thanks for your reply.**
> > >
> > > We sincerely thank you for appreciating our efforts in responding. The details with more experimental results and comparisons will be included in the final manuscript.

---

### Official Review · Reviewer_6wRC · 2023-07-12

**Soundness:** 2 fair
**Presentation:** 3 good
**Contribution:** 2 fair
**Rating:** 5
**Confidence:** 4

**Summary:**

The paper presents a novel method for generating 3D molecules that bind to specific protein pockets based on a functional-group-based diffusion model (D3FG). The model decomposes molecules into functional groups and linkers and generates their types, positions, and orientations gradually through a denoising process. The model uses equivariant graph neural networks to parameterize the denoisers and ensure the roto-translational invariance of the molecule distribution. The paper also introduces a new task of molecule elaboration, which aims to modify existing molecules based on the fragment hotspot maps of the protein pockets. The paper evaluates the model on the CrossDocked2020 dataset and shows that it can generate molecules with realistic structures, competitive binding affinity, and good drug properties. The paper also demonstrates that the model can perform molecule elaboration and generate molecules with higher affinity than the reference molecules. The paper claims that D3FG is a novel and effective method for structure-based drug design that leverages the pharmacological information of functional groups.

**Strengths:**

The paper proposes a functional-group-based diffusion model for pocket-specific molecule generation and elaboration called D3FG. The method decomposes molecules into two categories of components: functional groups defined as rigid bodies and linkers as mass points. The two kinds of components can form complicated fragments that enhance ligand-protein interactions. In the experiments, the authors claim the method can generate molecules with more realistic 3D structures, competitive affinities toward the protein targets, and better drug properties. The paper is original in its approach to generating molecules given the pockets’ structures of target proteins and its use of functional groups as basic components instead of atoms. The paper is clear in its description of the method and its results.

The paper is well-written and clearly describes the proposed method, its implementation details, and its evaluation results. The authors provide sufficient background information on related work and explain how their method differs from prior methods. The authors also provide detailed explanations of the model's components, such as the functional group decomposition, equivariant graph neural networks, and molecule elaboration. The authors use appropriate visualizations to illustrate their method's outputs and compare them with reference molecules.

The paper's originality lies in its novel approach to generating 3D molecules that can bind to specific protein pockets by leveraging functional groups' pharmacological information. The authors show their method can generate molecules with realistic structures, competitive binding affinity, and good drug properties. The paper's significance lies in its potential to improve structure-based drug design by enabling the generation of novel molecules that can bind to specific protein pockets with high affinity.

**Weaknesses:**

The paper's main weakness is that it does not provide a detailed comparison with prior methods for generating 3D molecules that can bind to specific protein pockets. The authors briefly mention some related work, but they do not provide a comprehensive comparison of their method with prior methods in terms of performance, efficiency, and scalability. The authors also do not provide a detailed analysis of the limitations of their method, such as the types of molecules it may not be able to generate or the types of protein pockets that it may not be able to bind to.

Another weakness of the paper is that it does not provide a detailed analysis of the interpretability and explainability of its method. The authors briefly mention some visualizations and explanations of their method's outputs. Still, they do not provide a systematic analysis of how their method's components contribute to its performance and how they can be interpreted in terms of pharmacological properties.

To improve the paper, the authors could perform a more comprehensive comparison with prior methods for generating 3D molecules that can bind to specific protein pockets, including both quantitative and qualitative analyses. The authors could also perform a more detailed analysis of the limitations of their method and how they can be addressed in future work. Finally, the authors could perform a more systematic analysis of the interpretability and explainability of their method, including sensitivity analyses, feature importance analyses, and pharmacophore analyses.

**Questions:**

- How did the authors select the 25 functional groups used for their method? How did they ensure these functional groups cover a diverse and representative range of pharmacological properties and structures? How sensitive is their method to the choice of functional groups?
- How did the authors evaluate the quality and diversity of the generated molecules? Did they use any metrics or criteria to measure the novelty, validity, and diversity of the generated molecules? How did they compare their method with prior methods regarding these metrics or criteria?
- How did the authors handle the cases where the generated molecules violate chemical or physical constraints, such as bond angles, bond lengths, steric clashes, or chirality? How did they ensure that the generated molecules were chemically feasible and stable?
- How did the authors handle the cases where the target protein pockets have multiple binding sites or modes? How did they ensure that their method can generate molecules that can bind to different sites or modes of the same protein pocket?
- How scalable is their method to larger, more complex protein pockets and molecules? What are the computational and memory requirements of their method? How does their method compare with prior methods regarding efficiency and scalability?

**Limitations:**

The authors have not adequately addressed the limitations and potential negative societal impact of their work. The authors only briefly mention some limitations of their method in the conclusion section, but they do not provide a detailed discussion of how these limitations affect their results and how they can be overcome in future work. The authors also do not discuss any potential negative societal impact of their work, such as the ethical, legal, or environmental implications of generating novel molecules that can bind to specific protein pockets.

---

> ### Author Rebuttal · Authors · 2023-08-06
>
>
> We thank you a lot for your constructive advice and answer your questions one by one, as below.
>
> Response to Q1:
> We use EFGS[1] to decompose the molecules, and analysis the stability of the substructures, so that manually established a repository of functional groups that most molecules can be decomposed in. As you can see in Table 8 in the Appendix, the `c1ccc2[nH]ccc2c1’ functional group is the 25th ones, only 2.3% of molecules have the substructures, so we assure the 25 functional groups are enough to consist of a repository and cover most generation tasks in Crossdocked. In detail, we test the sensitivity of performance affected by repository size in CQ2 in General response. We hope it can remove your doubts.
>
> Response to Q2:
> In D3FG, we did not compare these metrics except 'Validity' since we follow the protocol of experiments [2,3], which are different from tasks of molecule generation[4]. Here we conduct experiments on D3FG, and give these metrics on DiffSBDD and D3FG.
> 'Validity' is calculated as the ratio of generated 3D molecules that are chemically valid; 'Novelty' is defined in [5], where $C$ is the training set; 'Diversity' is the average pairwise Tanimoto distances. Table. 1 gives details.
>
> Table.1. molecule metrics of three diffusion methods.
> |            | Validity | Novelty | Diversity |
> |------------|----------|---------|------------|
> | DiffSBDD   | 95.61%   | 99.98%  | 0.704      |
> | TargetDiff | 95.73%   | 97.42%  | 0.718      |
> | D3FG       | 96.64%   | 96.81%  | 0.684      |
>
> The difference is minor, and these three models all perform well in these three metrics. The novelty and diversity of D3FG are relatively low because the substructures are fixed and the same as training and test data, so in Tanimoto distance calculation, some fingerprints are likely to be the same.
>
>
> Response to Q3: We follow DiffSBDD's post-process, by using Openbable to connect the atoms as molecules, and select the largest fragments when there are disconnected atoms. In this way, two metrics are relevant: connectivity and validity. The validity has been reported, and the connectivity is 87.16% of D3FG v.s. 79.52% of DiffSBDD reported in [6].
>
> Response to Q4: The Crossdocked datasets is a paired pocket-ligand datasets, so usually, previous methods (3DSBDD, Pocket2Mol, DiffSBDD, TargetDiff) all focus on 'one-to-one' task, and it cannot be generalized to 'one-to-many' or 'many-to-one' tasks. The molecules are generated by D3FG conditioned on a certain pocket structures, so when it is 'many-to-one' tasks, D3FG will use many pockets contextual information as inputs, and generate the corresponding number of molecules. We cannot ensure that the generated molecules can bind to a set of pockets if we do not know their structures and other information.
>
> Response to Q5:
> We here report the average generation time and memory usage of different diffusion methods on test set per 100 samples, and training time and memory per epoch. The batch size is set as 16 in training.
>
> |            | Time(Gen) | Memory(Gen) | Time(Train) | Memory(Trian) |
> |------------|-----------|-------------|-------------|---------------|
> | DiffSBDD   | 5'48"     | 4396MB      | 7'36"       | 31154Mb       |
> | TargetDiff | 15'52"    | 8944MB      | 34'43"      | 44138MB       |
> | D3FG       | 4'44"     | 4558MB      | 15'15"      | 31432MB       |
>
> It shows that TargetDiff is the most computationally comprehensive because the nodes in GNNs in TargetDiff are atoms (412.14 on average),  and DiffSBDD and D3FG are of no significant differences since the nodes in GNNs are amino acids in protein and atom/functional groups in DiffSBDD(68.10 nodes on average) and D3FG(53.62 nodes on average). Besides, on scalability, D3FG and DiffSBDD can handle most protein pockets, since the number of animo acids in a single pocket hardly exceeds 100.
>
> [1] Extended Functional Groups (EFG): An Efficient Set for Chemical Characterization and Structure-Activity Relationship Studies of Chemical Compounds. 2015
>
> [2] Pocket2Mol: Efficient Molecular Sampling Based on 3D Protein Pockets. 2022
>
> [3] 3D Equivariant Diffusion for Target-Aware Molecule Generation and Affinity Prediction. 2022
>
> [4] E(n) Equivariant Normalizing Flows. 2021
>
> [5] Graphvae: Towards generation of small graphs using variational autoencoders. 2018
>
> [6] Structure-based Drug Design with Equivariant Diffusion Models. 2022

---

> > ### Author Response · Authors · 2023-08-20
> > **Response.**
> >
> > Thank you for your valuable time and constructive suggestions.
> >
> > Considering that the author-reviewer discussion phase is nearing the end, we would like to be able to confirm whether our responses have addressed your concerns.
> >
> > If there is anything else that is not clear, please feel free to contact us.
> >
> > Best,
> >
> > Authors

---

### Author Rebuttal · Authors · 2023-08-06

**General Response**:

Here we conclude several common concerns of the reviewers, and respond to them as below:

CQ1: Sensitive Analysis (How the size of functional group repository affects the generative performance?)

We conduct experiments about the effects of the size of the repository on performance. Here, we choose top-5, top-10,... , top-20, top-25 functional groups in Table.8 in the paper as the smaller repository, and give Vina docking scores, Vina Delta affinity, and other four chemical properties in these situations. In the implementation, we add $-1e8$ to the logits corresponding to the removed 20, 15, 10, ... functional group types, to force the model to generate the remaining ones. Table.1 ( or Figure. 5 and Table. 11 ) in the newly updated supplementary material gives details, which will be added in the new version.

Table 1.  Effects of number of functional groups on performance
| repository size | 5      | 10     | 15     | 20     | 25     |
|-----------------|--------|--------|--------|--------|--------|
| Vina score      | -6.59  | -6.7   | -6.99  | -7.06  | -7.04  |
| Vina delta affinity  | 32.14% | 39.39% | 45.42% | 47.33% | 46.58% |
| qed             | 0.489  | 0.484  | 0.502  | 0.496  | 0.501  |
| sa              | 0.821  | 0.814  | 0.836  | 0.843  | 0.84   |
| logp            | 2.774  | 2.795  | 2.802  | 2.759  | 2.821  |
| Lipinski        | 4.91   | 4.983  | 4.937  | 4.931  | 4.965  |

It shows that in docking score, when the removed functional groups are small, the performance deterioration is minor, but significant when only 5 or 10 functional groups remains. One of the explanations is that the least common 5 to 15 functional groups just appear very rarely in the training set. For example, 'O = P(O)O', although it is the 15th most common functional group, occurs only 4167/100000 = 4.167%. Therefore, its exclusion will not have a particularly large impact on the overall performance. In contrast, when 'NS(=O)=O', which has a frequency of up to 10%, is excluded, its generation can only rely on the diffusion of linker atoms, which drastically reduces the frequency of occurrence and affects the overall performance. Besides, in QED, SA, and other scores, the differences are not significant.


CQ2: Detailed analysis of geometries like bond angle and dihedral.

We add some experimental results on JSD of bond angle and dihedral of the molecules generated by different methods v.s. references, as below. Table 2 and 3 demonstrate that D3FG generates more realistic drug molecules in comparison with other baselines.

Table 2. JSD of bond angle distributions.
| Angle   | Pocket2Mol | TargetDiff | DiffSBDD | D3FG(Stage) | D3FG(Joint) |
|---------|------------|------------|----------|-------------|-------------|
| C-C-C   | 0.269      | 0.272      | 0.304    | 0.255       | 0.253       |
| C-C-N   | 0.254      | 0.267      | 0.313    | 0.256       | 0.255       |
| C-N-C   | 0.286      | 0.241      | 0.319    | 0.269       | 0.277       |
| C-C-O   | 0.317      | 0.295      | 0.345    | 0.293       | 0.295       |
| C-O-C   | 0.308      | 0.311      | 0.372    | 0.310       | 0.304       |
| C-N-N   | 0.294      | 0.276      | 0.301    | 0.270       | 0.281       |
| N-C-O   | 0.300      | 0.295      | 0.326    | 0.282       | 0.291       |
| N-C-N   | 0.304      | 0.288      | 0.342    | 0.282       | 0.292       |


Table 3. JSD of dihedral distributions.
| Dihedral | Pocket2Mol | TargetDiff | DiffSBDD | D3FG(Stage) | D3FG(Joint) |
|----------|------------|------------|----------|-------------|-------------|
| C-C-C-C  | 0.151      | 0.149      | 0.158    | 0.141       | 0.138       |
| C-C-C-N  | 0.176      | 0.165      | 0.224    | 0.169       | 0.175       |
| C-C-C-O  | 0.183      | 0.159      | 0.206    | 0.156       | 0.164       |
| C-C-O-C  | 0.180      | 0.174      | 0.231    | 0.167       | 0.149       |
| C-C-N-C  | 0.165      | 0.142      | 0.223    | 0.136       | 0.146       |
| C-C-N-O  | 0.277      | 0.270      | 0.285    | 0.264       | 0.293       |
| C-N-C-O  | 0.453      | 0.430      | 0.398    | 0.358       | 0.335       |
| N-C-C-O  | 0.315      | 0.253      | 0.303    | 0.244       | 0.272       |
| C-N-C-N  | 0.340      | 0.317      | 0.328    | 0.254       | 0.263       |

CQ3: More related works on fragment-based molecule generation.

We add a paragraph to the related work in the new version, as below.

**Fragment-based drug design**. Previously, works on fragment-based molecule generation are proposed. For example, JT-VAE[44] generates a tree-structured scaffold over chemical substructures and combines them into a 2D-molecule. PS-VAE[45] can automatically discover frequent principal subgraphs from the dataset, and assemble generated subgraphs as the final output molecule in 2D. Further, DeepFrag [46] predicts fragments conditioned on parents and the pockets, SQUID[47] generates molecules in a fragment level conditioned on molecule’s shapes. FLAG[48] auto-regressively generates fragments as motifs based on the protein structures in 3D.

[44] Wengong Jin, Regina Barzilay, and Tommi Jaakkola. Junction tree variational autoencoder for molecular graph generation, 2019

[45]Xiangzhe Kong, Wenbing Huang, Zhixing Tan, and Yang Liu. Molecule generation by principal subgraph mining and assembling, 2022.

[46]Harrison Green, David R. Koesb, and Jacob D. Durrant. Deepfrag: a deep convolutional neural network for fragment-based lead optimization. Chemical Science, 2021

[47] Keir Adams and Connor W. Coley. Equivariant shape-conditioned generation of 3d molecules for ligand-based drug design. In The Eleventh International Conference on Learning Representations, 2023.

[48]ZAIXI ZHANG, Shuxin Zheng, Yaosen Min, and Qi Liu. Molecule generation for target protein binding with structural motifs. In International Conference on Learning Representations, 2023

---

### Decision · Program_Chairs · 2023-09-21

**Decision:**

Accept (poster)

**Comment:**

The paper presents a novel method for generating 3D molecules that bind to specific protein pockets based on a functional-group-based diffusion model (D3FG). The method is novel, the results are strong on QED and SA, and the new dataset would be beneficial for the community. Reviewers have major concerns about comparison with prior 3D fragment based approaches. Additional results provided in response are convincing. I would encourage authors to add these to the paper. Overall, I would support the paper to be included in the conference.